# The role of ATP-binding Cassette subfamily B member 6 in the inner ear

Stefanie A. Baril [1], Katie A. Wilson [2,3], Md Munan Shaik[4], Yu Fukuda [1], Robyn A. Umans[5], Alessandro Barbieri[6,7], John Lynch[1], Tomoka Gose [1], Alexander Myasnikov [4], Michael L. Oldham[4], Yao Wang[1], Jingwen Zhu[1,8], Jie Fang[9,10], Jian Zuo [9], Ravi C. Kalathur [4], Robert C. Ford[6], Allison Coffin[11], Michael R. Taylor [5,12], Megan L. O'Mara [2,13] & John D. Schuetz [1] ✉

ABCB6 has been implicated in dyschromatosis universalis hereditaria, a condition characterized by hyperpigmented and hypopigmented skin macules. Dyschromatosis universalis hereditaria can also present with hearing loss. Dyschromatosis universalis hereditaria-associated mutations in ABCB6 have been reported, but the role of this protein in the inner ear has not been studied. Here we determine a high-resolution (2.93 Å) cryo-EM structure of ABCB6 and functionally characterized several dyschromatosis universalis hereditaria mutants. We find that the L356P mutant abolishes ABCB6 function, and affirm the underlying loss of ATP binding mechanism using molecular dynamics simulations based on our cryo-EM structure. To test the role of ABCB6 in the inner ear, we characterize Abcb6 (the ABCB6 homolog) in zebrafish. We show that Abcb6 suppression by morpholinos reduces inner ear and lateral line hair cell numbers. Morphants also lack the utricular otolith, which is associated with vestibular function. Co-injecting morpholinos with human ABCB6 mRNA partially rescues the morphant phenotype, suggesting that Abcb6 plays a developmental role in inner ear structures. Further, we show that Abcb6 knockout mice exhibit an increased auditory brainstem response threshold, resulting in reduced hearing sensitivity. Taken together, these data suggest ABCB6 plays a role in inner ear development and function.

ATP-Binding Cassette (ABC) transporters are a diverse protein superfamily named for its members that use ATP hydrolysis to move substrates across lipid membranes[1,2]. All functional mammalian ABC transporters contain two highly conserved nucleotide binding domains (NBDs) along with at least two more varied transmembrane domains (TMDs), likely reflecting wide-ranging ABC transporter substrates that vary from simple ions to complex macromolecules. The ABC transporter superfamily is diverse with well-known roles in

[1]Department of Pharmaceutical Sciences, St. Jude Children's Research Hospital, Memphis, TN, USA. [2]Research School of Chemistry, College of Science, The Australian National University, Canberra, ACT, Australia. [3]Department of Biochemistry, Memorial University of Newfoundland, St. John's, Newfoundland and Labrador, Canada. [4]Department of Structural Biology, St. Jude Children's Research Hospital, Memphis, TN, USA. [5]Department of Chemical Biology and Therapeutics, St. Jude Children's Research Hospital, Memphis, TN, USA. [6]School of Biological Sciences, Faculty of Biology Medicine and Health, The University of Manchester, Manchester, UK. [7]Bioinformatics Institute (BII), Agency for Science, Technology, and Research (A*STAR), Singapore, Singapore. [8]Department of Pharmaceutical Sciences, University of Tennessee Health Science Center, Memphis, TN, USA. [9]Department of Developmental Neurobiology, St. Jude Children's Research Hospital, Memphis, TN, USA. [10]Department of Surgery, St. Jude Children's Research Hospital, Memphis, TN, USA. [11]Department of Integrative Physiology and Neuroscience, Washington State University Vancouver, Vancouver, WA, USA. [12]School of Pharmacy, Pharmaceutical Sciences Division, University of Wisconsin-Madison, Madison, WI, USA. [13]Australian Institute for Bioengineering and Nanotechnology, The University of Queensland, St. Lucia, Queensland, Australia. ✉e-mail: John.Schuetz@STJUDE.ORG

multidrug resistance as well as diseases and disorders. One super-family member, human ATP-binding cassette sub-family B member 6 (ABCB6), has been implicated in several seemingly disparate diseases[3–10]. ABCB6 is expressed ubiquitously as a half transporter which homodimerizes to form a functional transporter. ABCB6 is a porphyrin transporter that is abundant in erythroid cells[11], where it is capable of accelerating porphyrin biosynthesis and protecting against porphyrin overload[3–5,11].

With its high erythroid expression, the link between ABCB6 and blood-related disorders is not surprising. The absence of ABCB6 on red blood cell (RBC) membranes is the hallmark of the Langereis blood group[3]. ABCB6 was revealed as a modifier of porphyria, a disease of defective heme synthesis, using whole exome sequencing of patients with well-characterized porphyria disorders, coupled with pathway analysis and a genetic porphyria mouse model[4]. ABCB6 has also been linked to familial pseudohyperkalemia, a disorder where RBCs leak potassium at low temperatures[5]. However, assigning ABCB6 a role in a given disease is not always straightforward. For instance, ocular colo-boma, a developmental defect where tissue is missing from the eye, has also been linked to ABCB6, although phenotypic rescue experiments were not performed and ABCB6 knockdown was not validated[6]. Another ABCB6-linked disorder is dyschromatosis universalis heredi-taria (DUH), a rare heterogenous pigmentary genodermatosis, prominently characterized by alternating hyperpigmented and hypopigmented macules on the skin[7–10]. While the DUH macules are benign, other reported DUH symptoms include learning difficulties, photosensitivity, small stature, mental retardation, insulin-dependent diabetes mellitus, abnormalities in erythrocyte, platelet and trypto-phan metabolism, ocular abnormalities, and high-frequency hearing loss, which also occurs in some other epithelial pigmentation disorders[8,12].

First identified in 1933 in Japan by Ichikawa and Hiraga, DUH is commonly observed as an autosomal dominant disorder among families of Asian origin, although autosomal recessive or sporadic cases have been observed[8,10]. DUH has been divided into three sub-types, with DUH3 (OMIM 615402) corresponding to ABCB6-linked DUH[13]. Genome-wide linkage analysis was performed on two multi-generational DUH families, and two ABCB6 missense mutations, A453V and S322R, were found among affected individuals, identifying ABCB6 as a causative gene for DUH[9]. However, the latter mutation was first reported mistakenly as S322K before correct identification as S322R by another group[8,9]. To study protein function, these authors knocked down the ABCB6 ortholog in zebrafish (Abcb6), observing more mature melanocytes in the uninjected controls than in morphants. The authors did not comment on other DUH-associated symptoms (ocular defects, hearing loss, etc.) in their zebrafish morphants[8]. It should be noted that complete knockout of *Abcb6* in a pigmented C57BL/6 mouse strain resulted in no detectable abnormalities in pigmentation, eye morphology, or balance[14]. Therefore, it is conceivable that mutant ABCB6 proteins promote some of the DUH phenotypes by dysfunc-tional ABCB6 affecting protein quality control.

Since the link between ABCB6 and DUH was first established, sequencing studies have identified additional missense mutations, including S170G, L356P, Y424H, Q555K, G579E, T637A and G382R[7–10,15,16]. However, the impact of DUH mutations has not been investigated in-depth and no attempt has been made to determine if ABCB6 is related to the non-dermal symptoms of DUH. Several cryo-EM structures of ABCB6 have been reported[17–19], and here we report an additional high-resolution structure (2.93 Å) of human ABCB6. With this knowledge of ABCB6 structure, we sought to further probe ABCB6 DUH mutations using molecular dynamic (MD) simulations to com-plement biochemical assays, and studies in two tractable model organisms. In our examination of these DUH mutations, we uncovered unexpected evidence that ABCB6 may play a role in hearing loss, a common phenotype in genodermatoses like DUH[12].

## Results

### Expression of DUH mutants

Seven of the ten reported DUH missense mutations were chosen for study: S170G, S322R, L356P, Y424H, A453V, Q555K, and G579E (Fig. 1a). The incorrectly identified S322K mutation was excluded and the T673A and G382R mutations had not yet been reported at the start of our study[7–10,15,16]. All seven chosen mutations occur in highly conserved regions of ABCB6, suggesting a key role in ABCB6 function (Fig. 1a). To assess if expression or function was altered, we used multiple software prediction programs (Polyphen2[20], SiFT[21], and Mutation Accessor[22]). There was no consensus on the effect of each of these DUH mutations (Supplementary Table 1), therefore each DUH mutation was intro-duced into an ABCB6 expression vector and then transiently expressed in HEK 293 cells. For the majority of the DUH mutations, ABCB6 expression was comparable to ABCB6 WT, except for the S170G, S322R and L356P mutations, which showed significantly reduced expression (Fig. 1b, c). Several DUH mutants were able to bind ATP as evidenced by ATP-agarose pulldown (Fig. 1d, f), which was expected given their location outside the nucleotide binding domain. However, L356P binding to ATP-agarose could not be detected (Fig. 1d, f). To assess potential substrate interaction, we evaluated the mutants for binding to hemin-agarose beads. All the DUH mutants bound hemin-agarose beads comparably to ABCB6 WT (Fig. 1e, f). L356P was the only DUH mutation consistently predicted to have a negative effect on ABCB6 that also exhibited a significantly decreased expression level and ATP-agarose binding affinity (Supplementary Table 1, Fig. 1 b–d, f), and so we focused on this mutant for further studies.

### L356P mutant is stabilized by 4-Phenylbutyrate, a chemical chaperone

To increase L356P expression, HEK cells transiently expressing the L356P mutant were treated with one of three different compounds: MG132, a proteasome inhibitor, 4-phenylbutyrate (4-PBA), a chemical chaperone, and chloroquine (CQ), a lysosomal inhibitor[23]. MG132 and CQ treatment showed no significant difference between ABCB6 WT and the L356P mutant (Fig. 2a, b). 4-PBA treatment significantly increased L356P expression compared to ABCB6 WT (Fig. 2a, b). 4-PBA has been demonstrated to improve trafficking and chloride channel activity of the misfolded ΔF508 cystic fibrosis transmembrane con-ductance regulator (CFTR) mutant, another ABC transporter[24–26]. Additionally, 4-PBA prevented Parkin-associated endothelium receptor-like receptor (Pael-R) overexpression-induced aggregation, a hallmark of autosomal recessive juvenile parkinsonism[24,27]. The cha-perone activity of 4-PBA, established in other protein misfolding dis-eases, suggests L356P may be misfolded compared to ABCB6 WT[23,24].

### L356P is unable to interact with or hydrolyze ATP

After optimizing L356P expression conditions, ABCB6 L356P-Flag was expressed in the presence of 4-PBA to increase yield. ABCB6 E752Q-Flag, a catalytically inactive mutant, and ABCB6 WT-Flag were also expressed. All three proteins were purified by affinity chromatography followed by size-exclusion chromatography (Supplementary Fig. 1a, b). L356P showed negligible ATPase activity compared to ABCB6 WT (Fig. 2c), comparable to E752Q, the catalytically inactive mutant. Since L356P could not hydrolyze ATP, we next tested L356P for thermal stabilization by AMP-PNP, a non-hydrolysable ATP analog, to see if the L356P affected only ATP hydrolysis or both ATP binding and hydro-lysis. First, ABCB6 western blot signals were measured after incubation at increasing temperatures to empirically determine the melting temperature of both the ABCB6 WT and L356P mutant (Supplemen-tary Fig. 1e). Interestingly, the melting temperature of L356P ($T_m$ = 57.8 °C) was increased by 15.3 °C compared to ABCB6 WT ($T_m$ = 42.5 °C) (Supplementary Fig. 1f). Once the melting temperature was determined, isothermal shift assays were performed using various concentrations of AMP-PNP. L356P showed only a minor increase in

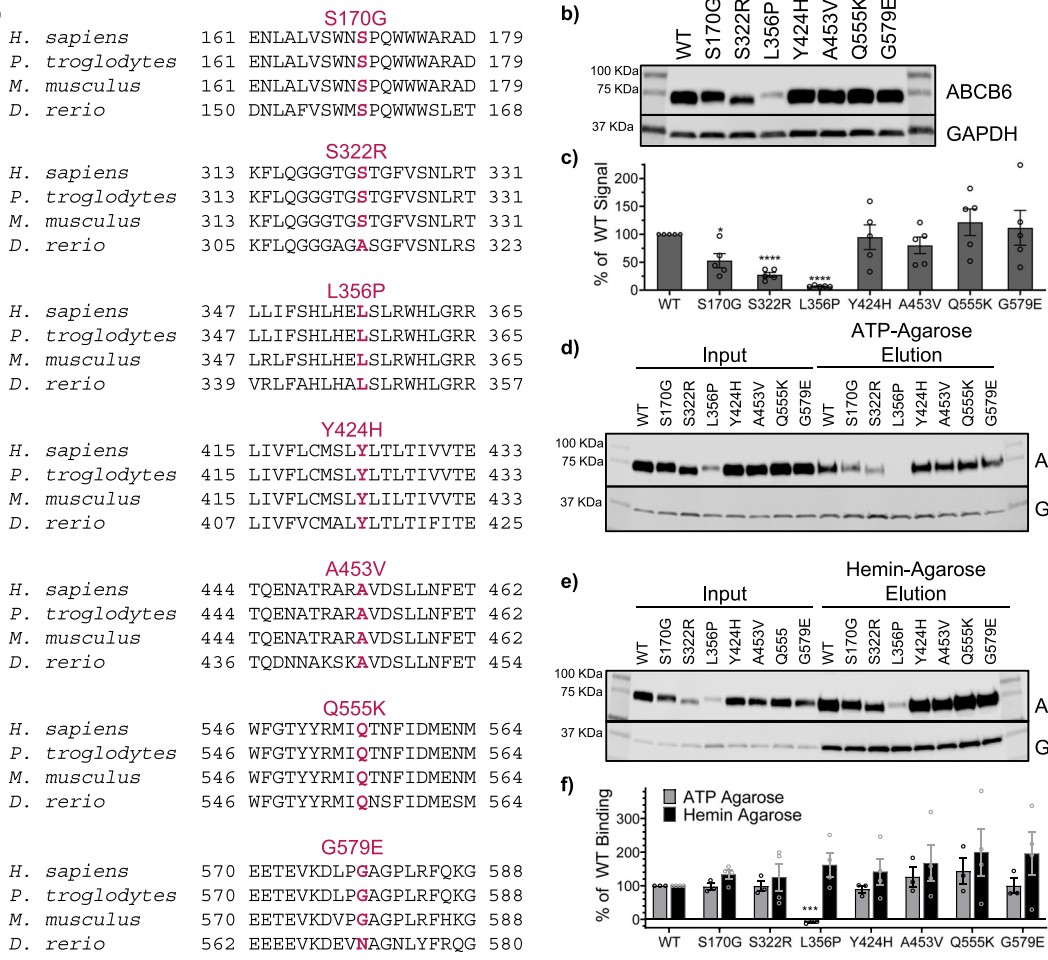

**Fig. 1 | ABCB6 variants identified in Dyschromatosis Universalis Hereditaria (DUH). a** DUH mutations occur in highly conserved regions of ABCB6, as shown in alignments with *Pan troglodytes* (chimpanzee, 99.5% identity), *Mus musculus* (mouse, 89.0% identity), and *Danio rerio* (zebrafish, 63.6% identity). All percent identities are compared to the human ABCB6. **b** Western blot shows most DUH mutations express similarly to ABCB6 WT (representative data shown). **c** Band signals from (**b**) and replicates show statistically significant reduction in expression level of S170G, S322R, and L356P ($p = 0.020$, < 0.0001, and < 0.0001, respectively, from unpaired two-tailed Welch's T-test compared to ABCB6 WT). Data reported as mean ± SEM, $N = 5$ biological replicates. Western blots show DUH mutants bind ATP-agarose (**d**) and hemin-agarose (**e**) comparably to ABCB6 WT. **f** Quantitation of

Western blot signals from ATP- and hemin-agarose pulldowns (**d**, **e**) and replicates show DUH mutants bind ATP-agarose and hemin-agarose comparably to ABCB6 WT. Percent of WT binding was calculated as described in Supplementary Methods. No signal of L356P could be detected in the ATP-agarose eluted fraction (**d**), but a faint band could be detected from the hemin-agarose elution fraction (**e**). No difference compared to ABCB6 WT was found by unpaired two-tailed Welch's T-test for any of the mutants binding to hemin-agarose. L356P showed a statistically significant decrease in ATP-agarose binding ($p = 0.0004$). Data are reported as mean ± SEM. ATP agarose $N = 3$ biological replicates, hemin agarose $N = 4$ biological replicates. Each biological replicate was repeated from transfection to western blot. Source data are provided in the Source Data file.

thermal stability with AMP-PNP compared to ABCB6 WT, suggesting that the L356P mutant poorly interacts with the nucleotide (Fig. 2d, e).

**Structural insights into L356P from Cryo-EM and MD simulations**

At the start of our investigation, the structure of ABCB6 had not yet been reported and models of ABCB6 WT were first predicted using the I-TASSER server[28]. We purified ABCB6 in the presence of n-dodecyl-β-maltoside (DDM) / cholesteryl hemisuccinate (CHS) and obtained a reconstruction of ABCB6 to a resolution of 3.5 Å. However, once other structures were published, the microscopy data was re-processed using the atomic coordinates of ABCB6 WT (PDB ID: 7DNY)[17], resulting in a reconstruction of ABCB6 to a resolution of 2.93 Å, helping to solve some ambiguity of in the NBD-TMD interface region surrounding L356 in the NBD coupling helix (EMDB ID: EMD-46724, PDB ID: 9DBQ), Fig. 2f, g; Supplementary Table 2, Supplementary Figs. 2–4). Our final model is in good agreement with other deposited ABCB6 structures in the inward-facing conformation[17–19]. Among these, three

ABCB6 structures (PDB IDs 7D7N[19], 7D7R[19], and 7EKM[18], resolved to 5.20 Å, 4.00 Å, and 3.60 Å, respectively) are representative of the apo ABCB6 WT transporter and all share a high similarity with our cryo-EM dataset (PDB ID: 9DBQ; heavy atom RMSD between 2.4 and 2.9 Å). Notably, the resolution of our cryo-EM structure (2.93 Å) is higher than previous cryo-EM structures. In addition to the canonical TMD and NBD, ABCB6 contains an additional N-terminal TMD termed $TMD_0$. At present, the role of the $TMD_0$ has not been well studied, although $TMD_0$ has been theorized to aid in protein-protein interactions or trafficking[29]. In most of the published structures in detergent and nanodiscs, including our own, density for the $TMD_0$ is missing from the final model. In our preparation of ABCB6 in DDM/CHS, the $TMD_0$ is too flexible to emerge in the 2D class projections. This observation is consistent with a previous structural determination of ABCB6 bound to hemin and glutathione (PDB ID: 7DNZ)[17] in the presence of 6-Cyclohexyl-1-Hexyl-β-D-Maltoside (CYMAL-6)/CHS, another detergent system[17]. In another previous 5.20 Å reconstruction of ABCB6 in the presence of 2,2-didecylpropane-1,3-bis-β-D-maltopyranoside (LMNG)

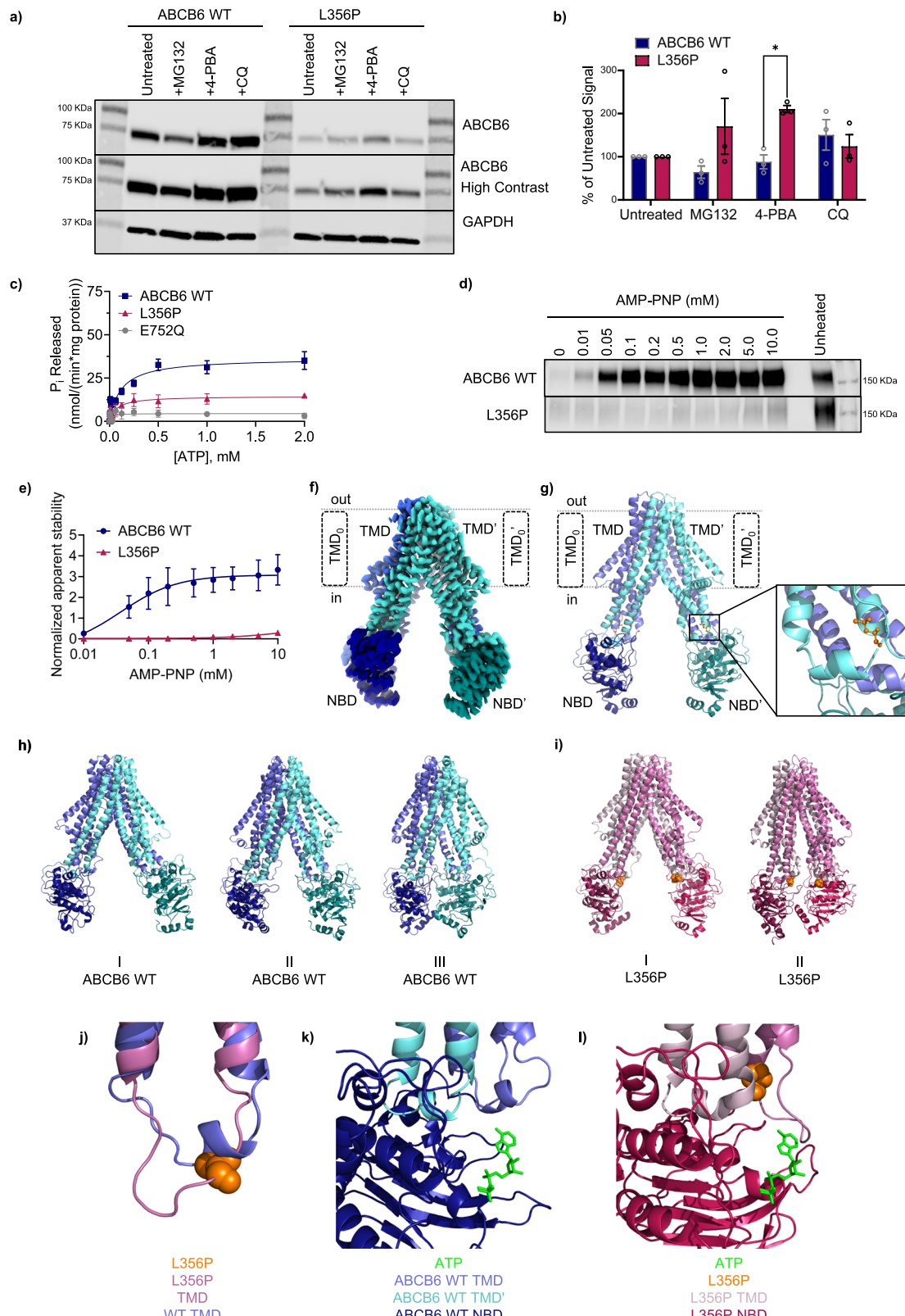

<table>
</table>

(PDB ID: 7D7N)[19], the TMD0 is evident in the 2D class projections with density in the final reconstruction, however the resulting structure is 5.2 Å[19]. We also observed the presence of the TMD0 in 2D class projections from a lower resolution reconstruction when using LMNG in the absence of CHS, but chose the higher-resolution structure resulting from the DDM/CHS detergent system. We prioritized the global resolution of the structure over the TMD0 so that the resulting structure would be appropriate for MD simulations.

Using our cryo-EM structure of ABCB6 WT, MD simulations were employed to study the effect of the L356P mutation on ABCB6 structure. Principle component analysis of the triplicate 500 ns MD simulations of both the apo state of ABCB6 WT and the L356P mutant indicated that the introduction of the L356P mutation altered the region of conformational space sampled by the L356P mutant relative to the ABCB6 WT protein (Supplementary Fig. 5a). Furthermore, when the structures adopted over the total simulation time were

**Fig. 2 | Characterization of the L356P mutation. a** Western blot shows L356P is disproportionately stabilized by 4-phenylbutyrate (4-PBA) compared to ABCB6 WT. MG132 and chloroquine (CQ) also show a less significant effect on expression (representative data shown). **b** Band signals from (a) and replicates were quantitated and the percentage of untreated signal was calculated (see Supplementary Methods). ABCB6 WT is shown in indigo, L356P is shown in magenta. Analysis using an unpaired two-tailed Welch's T-test and found no significant differences between ABCB6 WT and L356P except for with 4-PBA treatment ($p = 0.00262$, $N = 3$ biological replicates per treatment.) Data are reported as mean ± SEM. **c** L356P (magenta) has significantly decreased ATPase activity compared to ABCB6 WT (indigo). The catalytically inactive E752Q mutant is shown in grey. Data are reported as mean ± SEM, $N = 15$ experiments for ABCB6 WT from five biological replicates, $N = 8$ experiments for E752Q from three biological replicates, and $N = 4$ for L356P from two biological replicates. **d** Western blot of isothermal shift assay shows L356P exhibits little thermal stabilization with AMP-PNP treatment, suggesting L356P

cannot interact with ATP (representative data shown). **e** Band signals from (d) and replicates were quantitated. Data are reported as mean ± SEM. $N = 3$ technical replicates for ABCB6 WT (indigo) and L356P (magenta). **f** Cryo-EM structure of ABCB6 WT (EMDB ID: EMD-46724, PDB ID: 9DBQ) and resulting cartoon model (**g**). Nucleotide binding domains (NBDs) are shown in darker tones while transmembrane domains (TMDs) are colored in lighter tones. L356 (orange) is located near the coupling helix of ABCB6 (inset). Predominant conformations of ABCB6 WT (**h**) and L356P (**i**) over the combined 1500 ns of MD simulation as selected from hierarchical agglomerative clustering analysis (**j**) Overlay of ABCB6 WT (purple) and L356P (pink) showing the change in conformation of the coupling helix. A single chain of the homodimer has been shown for clarity, however similar changes are observed in both monomers. Overlay of ATP from ABCB10 (PDB ID: 4AYT) onto ABCB6 WT (**k**) or L356P (**l**) showing the change in conformation of the ATP binding site. Source data are provided in the Source Data file.

clustered, the ABCB6 WT spanned conformations that had a varying degree of separation between the NBDs, ranging from a fully separated conformation to a conformation with the NBDs in contact with one another (Fig. 2h). Conversely, the L356P mutant adopted conformations with the NBDs either in contact with one another or in close proximity (Fig. 2i). To gain further insight into the difference in rigidity of the ABCB6 WT and L356P mutant, the RMSD and RMSF were calculated. ABCB6 WT showed a slightly higher average heavy atom RMSD (6.8 Å vs 6.1 Å) and higher average RMSF over the trajectory compared to the L356P mutant (Supplementary Fig. 5b). The reduced conformation space explored by the L356P mutant relative to ABCB6 WT coupled with higher RMSD/RMSF for ABCB6 WT relative to the L356P mutant indicated that the L356P mutant may have an overall greater rigidity. The increased rigidity of the L356P mutant is consistent with the elevated $T_m$ of the L356P mutant since thermostability is associated with an enhanced rigidity of the protein backbone[30].

More locally, a hydrogen bond was formed between L356(NH) and His352(O) of ABCB6 WT for 55% of the simulation in Chain A and 63% of the simulation in Chain B. This observation is consistent with classic $i + 4$ hydrogen bonding of an alpha helix. Conversely, in the L356P mutant, backbone hydrogen bonding was lost due to conversion of the backbone amide to an imide, and no significant hydrogen bonds are formed throughout the simulations. Consistent with the loss of hydrogen bonding in the L356P mutant, defined secondary structure protein (DSSP) analysis indicated that the L356P point mutation led to complete loss of the helical character of this residue (Supplementary Fig. 5c, d), while L356 in ABCB6 WT exhibited 50% helix propensity (Supplementary Fig. 5c, d). As proline traditionally disrupts secondary structure, the loss of helical character upon introduction of the L356P point mutation causes premature termination of the coupling helix. This effect was observed in MD simulations of the L356P mutant, where an extended loop directed towards the NBD was formed (Fig. 2j). To investigate whether the formation of this extended loop impacted the binding of ATP to the NBD, ATP was overlayed onto a representative structure from the MD simulations, based on the ATP binding position of a homologous ABC transporter (ABCB10, PDB ID: 4AYT)[31]. The ATP binding site is highly conserved across ABC transporters and any mutations near the ATP binding site motifs are likely to perturb the dimerization process and decrease ATP binding affinity. ABCB6 WT displayed an open ATP binding site that is comparable to binding sites observed for other ABC transporters (Fig. 2k). Conversely, the elongated loop of the L356P mutant extended across the ATP binding site, occluding the site, and causing direct clashes with the overlayed binding conformation of ATP (Fig. 2l). These simulations predicted that the L356P mutant would not bind ATP due to steric hindrance, in accordance with the results of our ATP-agarose pull-downs (Fig. 1d, f) and thermal-shift assays with AMP-PNP (Fig. 2d, e). Hydrogen bonding analysis showed a reduction in hydrogen bonding for L356P relative to ABCB6 WT for hydrogen bonds involving the

highly conserved Walker A, Walker B, and Q-loop motifs (Supplementary Table 3), while no change in hydrogen bonding is observed for the conserved ABC signature motif and an increase in hydrogen bonding is observed for the H-loop motif. The changes in hydrogen bonding observed in the NBD indicate that the L356P mutation produces global effects on the structure of ABCB6 that will generate a nonfunctional heterodimer.

## Characterization of an ABCB6 homolog in zebrafish (Abcb6)

Given our data suggesting L356P was a nonfunctional mutant, we wanted to explore the possibility that ABCB6 dysfunction or knockdown plays a role in hearing. Because there were no current reports that ABCB6 mutations affected auditory or vestibular alterations, we explored the role of ABCB6 in the auditory/vestibular apparatus in zebrafish (*Danio rerio*), a common model organism in auditory research[12,32–34]. Zebrafish are often utilized in hearing research because several molecular mechanisms of inner ear development in zebrafish are conserved with mammals[35,36]. Despite the lack of a cochlea, the structure of the zebrafish inner ear resembles that of other vertebrates, containing multiple epithelia populated with sensory hair cells and surrounding supporting cells. Fishes also possess a second hair cell-bearing sensory system called the lateral line, where each sensory organ (neuromast) is comprised of clusters of hair cells and associated supporting cells[37,38]. Lateral line hair cells detect near-field water movement important for a variety of behaviors including predator avoidance, prey detection, orientation to current, and schooling[39].

The zebrafish *abcb6* gene (*abcb6a*, NCBI reference sequence NP_001139165.1) was identified, amplified, and sequenced from zebrafish cDNA. Although zebrafish have two *abcb6* genes (*abcb6a and abcb6b*), only *abcb6a* has been categorized as the human ABCB6 ortholog[40] based on the gene structure and sequence (Supplementary Fig. 6a-c). The zebrafish Abcb6 protein shares 61.4% identity and 74.6% similarity with the human ABCB6 homolog, although zebrafish Abcb6 lacks the sole, rare, "N-X-C" N-linked glycosylation site of many mammalian ABCB6 orthologs (Fig. 3a)[41,42]. Once zebrafish *abcb6a* was amplified from zebrafish cDNA and cloned into an expression vector, zebrafish Abcb6 glycosylation was probed with PNGase F treatment, which enzymatically removes N-linked oligosaccharides, irrespective of the maturation state of the glycoprotein[41]. As expected, the molecular weight of human ABCB6 shifts with PNGase F treatment, while the molecular weight of zebrafish Abcb6 remains unchanged by PNGase F treatment, confirming that the zebrafish homolog is not a glycoprotein (Fig. 3b). Although zebrafish Abcb6 is not a glycoprotein, both bind ATP, as evidenced by ATP-agarose pulldown (Fig. 3c) as well as a variety of porphyrins, as evidenced by hemin-agarose pulldown with porphyrin competition (Fig. 3d).

Our cryo-EM structure of human ABCB6 was used to develop a homology model of zebrafish Abcb6 and triplicate 500 ns MD

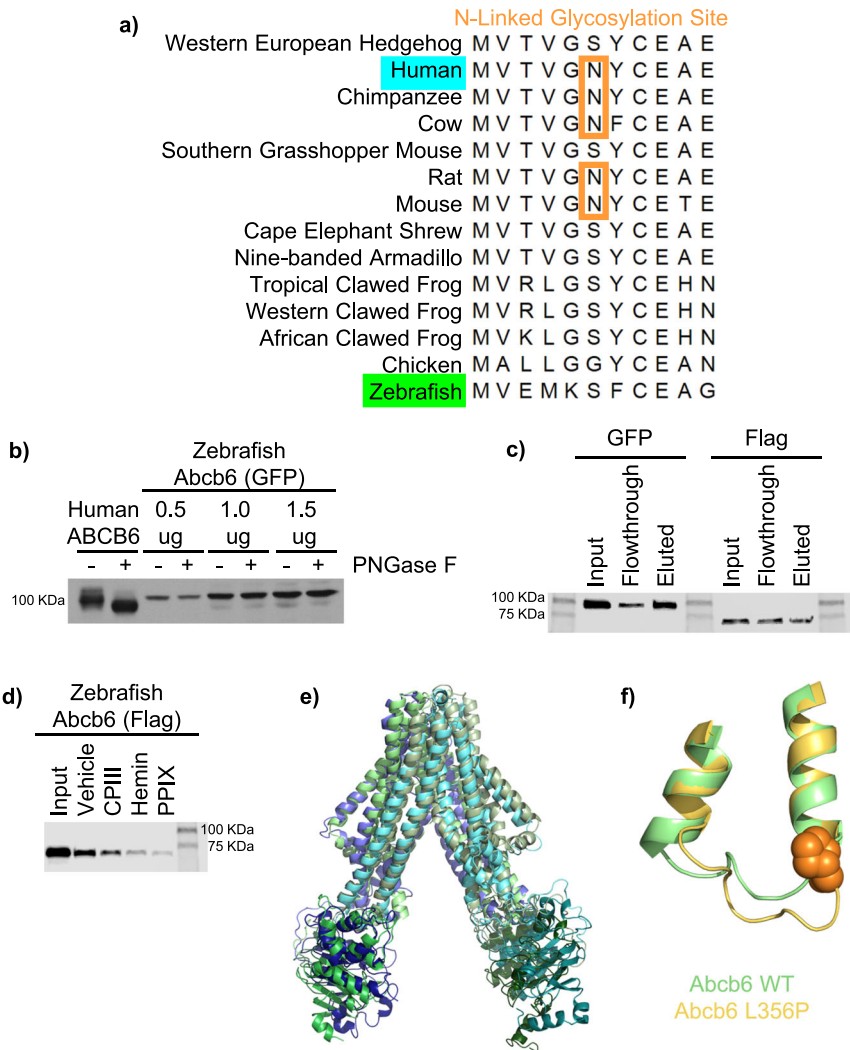

**Fig. 3 | Characterization of zebrafish ABCB6 (Abcb6). a** Alignment of homologous ABCB6 sequences shows zebrafish Abcb6 lacks the N-linked glycosylation site[89]. **b** Western blot of samples from PNGase F treatment confirms that zebrafish Abcb6 is not a glycoprotein, unlike its human homolog (*N* = 1 as experiment is a qualitative confirmation of sequencing data). Western blots show zebrafish Abcb6 binds ATP-agarose (**c**) and various porphyrins like its human homolog (**d**). Representative data shown, *N* = 3 biological replicates for each pulldown, with each experiment repeated from transfection to western blot. **e** Overlay of a representative structure from the MD simulations of the zebrafish Abcb6 (green) and ABCB6 WT (indigo/turquoise as shown in Fig. 2) showing that the two proteins adopt a similar conformation. **f** Overlay of zebrafish Abcb6 WT and zebrafish Abcb6 L356P showing the change in conformation of the coupling helix. Source data are provided in the Source Data file.

simulations were performed to investigate the structural dynamics of the zebrafish homolog. All domains retained similar conformations relative to the human ABCB6 structure (heavy atom RMSD = 1.8-2.7 Å) (Fig. 3e). Nevertheless, differences were seen in the relative orientation of the domains. Although conformational transitions can take place over milliseconds, during our triplicate 500 ns MD simulations, zebrafish Abcb6 exhibited no global conformational changes (Supplementary Fig. 7a). As observed for human ABCB6, the introduction of the L356P mutation into zebrafish Abcb6 leads to premature termination of the coupling helix proximal to the L356P mutation in simulations (Fig. 3f, Supplementary Fig. 7), indicating that the L356P mutation may have a similar effect in zebrafish and human ABCB6 proteins. The zebrafish Abcb6 homology model was also compared to the AlphaFold structure of zebrafish Abcb6 (accession code: F1QCK2)[43]. Superimposing the Abcb6 homology and AlphaFold models gives a backbone RMSD of 1.9 Å, indicating there is very little difference between the two. When the individual domains of Abcb6 are superimposed, these small structural differences are further decreased, evidenced by a backbone RMSD of 1.5 Å in the TMD, and

1.1 Å in each NBD. The largest structural deviation is observed in the orientation of the coupling helix in the TMD, nevertheless in simulations initiated from the homology model, both the AlphaFold and homology model conformations of the coupling helix were sampled (Supplementary Fig. 8).

### Zebrafish studies of Abcb6 function

Whole mount in situ hybridization (WISH) was used to visualize the sites of most abundant expression of Abcb6 in zebrafish larvae, with a focus on the inner ear. When treated with a zebrafish *abcb6* anti-sense mRNA probe, WISH revealed signal in the otic vesicle (Fig. 4a, i-iii) with no specific detectable signals in these locations for the control zebrafish *abcb6* sense probe (Fig. 4a, iv). To examine Abcb6 function in zebrafish, we designed splice-donor morpholinos to exon 15 (MO15) to knock down expression. Knockdown was confirmed by RT-PCR (Supplementary Fig. 9). At two different times post-fertilization (dpf), zebrafish Abcb6 MO15 treatment of embryos revealed that the number of hair cells per lateral line neuromast was decreased significantly irrespective of their anatomical location (head, trunk, tail) compared

to the WT uninjected control (Fig. 4b), suggesting zebrafish Abcb6 is required for proper lateral line development. Gene knockdowns using morpholinos can exhibit p53-mediated off-target effects[44]. Therefore, we conducted a control experiment using the zebrafish Abcb6 MO15 in p53-null zebrafish rather than the previously utilized AB zebrafish. There was no difference in lateral line hair cell number between fish injected with the Abcb6 MO15 alone and fish co-injected with the Abcb6 MO15 and p53 morpholinos (t-test, $p = 0.35$, Supplementary Fig. 10), suggesting that hair cell reduction was a specific result of zebrafish *abcb6* knockdown.

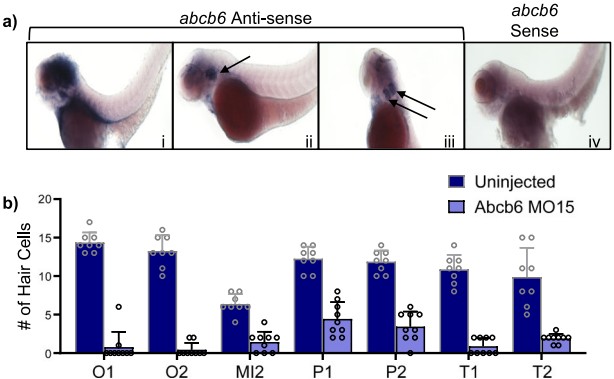

**Fig. 4 | Zebrafish abcb6 knockdown affects lateral line development in zebrafish. a** Whole mount in situ hybridization (WISH) of 3 dpf AB zebrafish shows Abcb6 expression in the inner ear when treated with a zebrafish abcb6 anti-sense mRNA probe (i-iii). The auditory vesicle is denoted with black arrows. Zebrafish Abcb6 expression is not detected with a zebrafish abcb6 sense mRNA probe (iv).
**b** Zebrafish Abcb6 MO15 morphants (purple) developed a reduced number of hair cells compared to WT (indigo) in zebrafish neuromasts. Data were analyzed by 2-way ANOVA with genotype and neuromast as factors. There is a significant main effect of genotype ($F_{1,111} = 126.6$, $p < 0.0001$) and neuromast ($F_{6,111} = 6.13$, $p < 0.0001$). O1, O2, and MI2 neuromasts are found on the head, P1 and P2 represent the first two neuromasts of the posterior lateral line (trunk), and T1 and T2 are the two terminal-most neuromasts on the tail (neuromast nomenclature modified from Raible and Kruse, 2000)[90]. Hair cell counts are from 3 dpf brn3c transgenic larvae, $N = 8$ fish for uninjected, $N = 9$ fish for Abcb6 MO15, bars represent mean + 1 SEM. The person counting hair cells was blinded to treatment. Experiment was repeated and the results showed the same pattern. Source data from (**b**) are provided in the Source Data file.

The larval zebrafish ear contains two hair cell-bearing epithelia, the saccule and utricle, each with an associated calcified otolith. Abcb6 MO15 injection also disrupted utricular otolith formation up to 14 dpf but did not alter saccular otolith formation (Fig. 5a), while the uninjected control larvae developed both the saccular and utricular otoliths. Abcb6 MO15 injection also significantly reduced the number of hair cells in the anterior inner ear epithelium, which is associated with the utricular otolith (Supplementary Fig. 10).

As with the lateral line experiment, we repeated the Abcb6 MO15 injections with p53-null background zebrafish[44]. Both the WT (AB background) and p53-null uninjected fish developed two normal otoliths per ear (Fig. 5b), while both the WT (AB background) and p53-null fish injected with the zebrafish Abcb6 MO15 failed to develop the utricular otolith. The absence of the utricular otoliths in the p53-null morphants indicated that the morphant phenotype was not due to p53-mediated off-target effects, similar to our result in the lateral line. Otolith development with Abcb6 MO15 treatment could be partially rescued by co-injection with human *ABCB6* mRNA (Fig. 5c, Supplementary Table 4). The majority (52.6%) of morphants co-injected with *ABCB6* mRNA developed both saccules and utricles (WT phenotype), while an additional 42.1% exhibited a partial phenotype with 2 saccular otoliths and one utricular otolith (loss of only one otolith). Only 5.3% of co-injected morphants retained the 2-saccule phenotype of the zebrafish Abcb6 morphant alone. Abcb6 morphants also exhibited abnormal swimming patterns, with a sizable portion of the morphants occupying the lower third of the tank (Fig. 6a-f). Although the difference between horizontal distance traveled was not statistically significant for the two groups (Fig. 6g), Abcb6 morphants traveled a significantly longer vertical distance than their uninjected counterparts (Fig. 6h), indicative of vestibular or lateral line dysfunction. Taken together, these data suggest that Abcb6 is required for normal inner ear development in zebrafish.

To confirm our initial results with MO15 injection, we designed additional morpholinos to another splice junction in exon 7 (MO7) and knockdown was confirmed by RT-PCR (Supplementary Fig. 9). For both morpholinos, a statistically significant decrease in hair cell number was observed (Supplementary Fig. 11). MO15 injection showed a dose-dependent effect on utricular development (Supplementary Fig. 12). However, the single otolith phenotype was not observed for MO7 morphants (Supplementary Fig. 12).

To connect our results in zebrafish back to mammals, the Shared Harvard Inner-Ear Laboratory (SHIELD) database was consulted[45]. Data

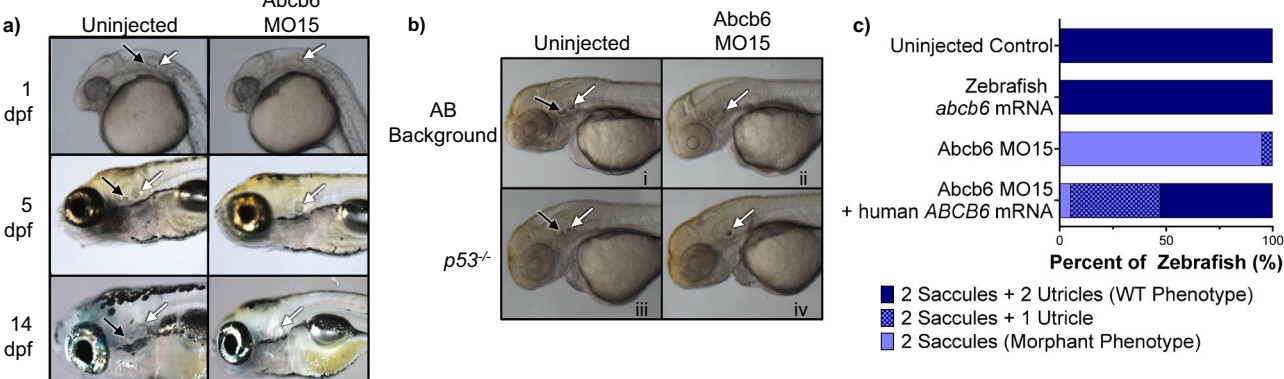

**Fig. 5 | Zebrafish Abcb6 MO15 morpholino affects otolith development in zebrafish. a** Zebrafish Abcb6 MO15 disrupts formation of the utricular otolith (black arrow), while uninjected controls retain both utricular (black arrow) and saccular (white arrow) otoliths at 1-, 5-, and 14-days post-fertilization (dpf).
**b** Uninjected wild-type fish with AB (i) and p53-null (iii) backgrounds develop normal otoliths, while Abcb6 morphants (ii and iv) fail to develop the utricular otolith. **c** The single otolith phenotype can be rescued by co-injection with human ABCB6 mRNA. Data for uninjected control ($N = 25$ fish), zebrafish abcb6 mRNA ($N = 9$ fish), Abcb6 MO15 ($N = 20$ fish), and Abcb6 MO15 + human ABCB6 mRNA ($N = 19$ fish) can be found in Supplementary Table 4 or the source data file. Injections were repeated and the results showed the same pattern.

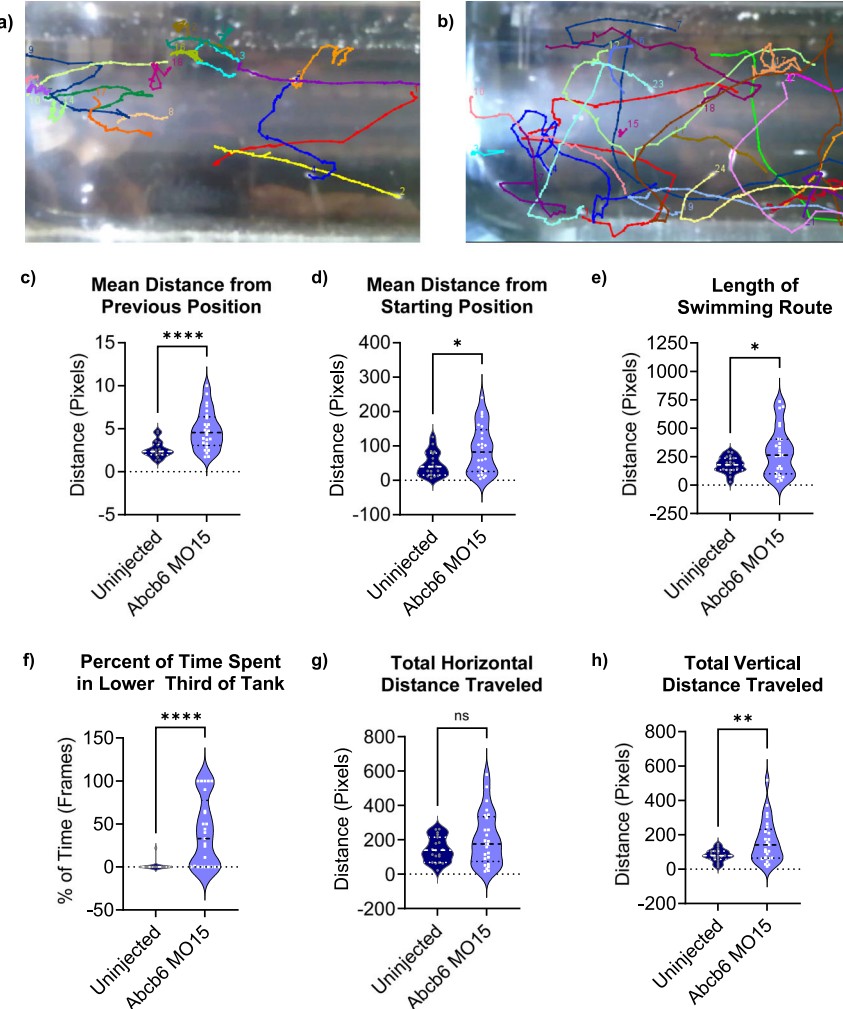

**Fig. 6 | Zebrafish Abcb6 MO15 morphants show altered behavior compared to uninjected zebrafish.** Zebrafish movement paths overlaid with last video frame shows uninjected zebrafish swim smoothly in the upper 2/3 of the tank (**a**), while the zebrafish Abcb6 MO15 morphants swim erratically and spend a significant amount of time in the lower third of the tank (**b**). Abcb6 MO15 morphants (purple) swam further with each video frame (**c**), $p = < 0.0001$, and swam farther from their starting position (**d**), $p = 0.013$, than the uninjected controls (indigo). Abcb6 MO15 morphants also swam longer routes than their uninjected counterparts (**e**), $p = 0.0176$, and spent a significantly higher percentage of time in the bottom of their tank (39.9%) compared to the uninjected control (1.2%) (**f**), $p = < 0.0001$. Although there was no significant difference in total horizontal distance traveled (**g**) $p = 0.0785$, Abcb6 MO15 morphants traveled a greater vertical distance (160.7 pixels) than their uninjected controls (79.7 pixels, (**h**), $p = 0.0041$. * = $p < 0.05$, ** = $p < 0.01$, **** = $p < 0.0001$, ns = not significant by unpaired two-tailed Welch's T-test compared to uninjected control. Data from 21 dpf AB zebrafish, N = 18 fish for uninjected, N = 25 fish for Abcb6 MO15. Median is shown as a dashed line and upper and lower quartiles are shown as dotted lines. Source data are provided in the Source Data file.

found in the SHIELD database showed mouse *Abcb6* mRNA counts in the cochlea (auditory organ) increased during early postnatal development (Fig. 7a). Conversely, mRNA counts in the utricle, which is important for balance, increased between E16 and P0 but did not show a linear increase during the first postnatal week. Several heme biosynthesis genes, such as ferrochelatase and mitoferrin, also showed increases during early development (Supplementary Fig. 13). To evaluate the effect of ABCB6 in the mammalian inner ear, auditory brainstem response (ABR), a gross physiological measure of hearing sensitivity, was measured in WT and *Abcb6* knockout (KO) mice. KO mice exhibited increased ABR thresholds at all frequencies tested compared to WT mice, suggesting reduced hearing sensitivity in *Abcb6* KO mice (Fig. 7b). RNA-Seq of cochlea isolated from 2–3-month-old WT and KO mice revealed significant changes in several genes implicated in hearing (Supplementary Fig. 14). Zinc finger and BTB domain containing protein 20 (Zbtb20), a protein essential for maturation of the cochlea and hearing in mice[46], was increased in *Abcb6* KO cochlea (Supplementary Fig. 14a). Additionally, potassium voltage-gated

channel subfamily A member 10 (Kcna10) and epidermal growth factor receptor (Egfr) expression were increased in *Abcb6* KO cochlea (Supplementary Fig. 14a). Kcna10 KO mice exhibit hearing loss[47] and EGF signaling has been implicated in cochlear development[48]. Sperm associated antigen 6 (Spag6), implicated in polarity defects in planar cells and hearing loss[49], and dynein cytoplasmic 1 intermediate chain 1 (Dync1li1), requisite for mammalian cochlear hair cell survival[50], showed decreased expression in *Abcb6* KO cochlea compared to WT cochlea (Supplementary Fig. 14b, c, respectively). Consistent with our zebrafish studies, these data from mice suggest ABCB6 is vital for development and function of the inner ear. However, it is difficult to theorize how the knockdown of ABCB6 causes the changes in hearing gene expression observed in the RNA-Seq data.

## Discussion
Our study expressed the majority of the DUH mutations in parallel and revealed structural and functional deficits. We evaluated seven DUH mutants but focused on L356P because the L356P expression level and

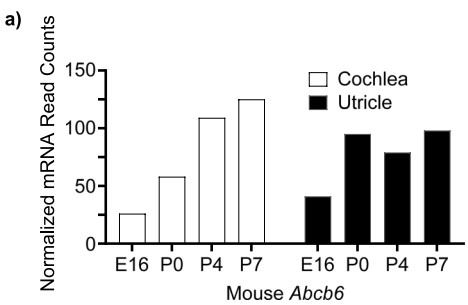
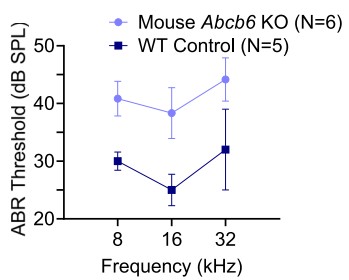

**Fig. 7 | ABCB6 plays a role in the murine inner ear. a** mRNA counts of mouse Abcb6 increase with cochlear development (white), but not utricular development (black). Data taken from the SHIELD database[45]. **b** 5–6-week-old Abcb6 knockout (KO) mice (purple) exhibit a frequency-independent increase in auditory brainstem response (ABR) compared to the wild-type control (indigo). Data are reported as mean ± SEM. $N = 5$ mice for WT Control (C57/129 mix) and $N = 6$ mice for Abcb6 KO. Source data are provided in the Source Data file.

interaction with ATP showed the greatest reduction (Fig. 1b–d, f). Prior to structural elucidation, the L356P mutation was not expected to interrupt function due to its location outside of the nucleotide binding domain. However, the position of L356P near the coupling helix (Fig. 2g) greatly affected the accessibility of the ATP-binding pocket, partially occluding the binding site and narrowing the ATP-accessible interface (Fig. 2j–l). MD simulations were initiated from a folded protein in which the L356P point mutation was introduced, therefore we do not know how the mutation affects the global folding of the L356P mutant. However, L356P was stabilized by 4-PBA (Fig. 2a, b; Supplementary Fig. 1d), a chemical chaperone previously demonstrated to improve trafficking and chloride channel activity of the misfolded ΔF508 CFTR mutant[24–26] and also to prevent Pael-R overexpression-induced aggregation, a hallmark of autosomal recessive juvenile parkinsonism[24,27]. The chaperone activity of 4-PBA suggests L356P may be misfolded compared to ABCB6 WT. Since ABCB6 requires ATP hydrolysis in both NBDs, and the coupling helix coordinates the NBD and TMD motion during substrate transport, it is highly likely that a ABCB6 WT/L356P heterodimer is nonfunctional. Therefore, L356P likely acts as dominant-negative-allele, as proposed previously by Zhang et al[10].

Although our studies focused on L356P, two other DUH mutations showed significantly decreased expression levels compared to ABCB6 WT. The S170G mutation is located in the $TMD_0$ domain and is likely located in an extracellular loop. Although it is difficult to hypothesize the effect of S170G given that the S170 residue is absent from all published ABCB6 structures, decreased expression of the S170G mutant suggests the serine residue is important for the stability of ABCB6. The S322R mutation is located in the "TM7 bulge loop" as described by Kim et al[17], comprising residues 318–324. Although Kim et al. knocked out several residues comprising the TM7 bulge loop, the expression level of these constructs was not discussed. However, Kim et al. theorized that complete deletion of the TM7 bulge loop desta-bilized the inward-facing conformation, and so it is possible that mutations in this loop may also destabilize the protein. Our pre-liminary characterizations of the other DUH mutations show that the mutants bind hemin-agarose and ATP-agarose similarly to ABCB6 WT. However, these mutants may behave differently than ABCB6 WT in their localization, transport of substrates, or hydrolysis of ATP, since binding does not necessarily correlate to hydrolysis or transport rates.

Interestingly, *ABCB6* (/) individuals from the Langereis blood group exhibit no obvious abnormalities, so it is puzzling that hetero-zygous DUH individuals show more of a phenotype than *ABCB6*-null individuals[3]. The majority of *ABCB6* (/) individuals (or Lan –) carry nonsense, frameshift, or splice junction mutations, causing global changes in the protein, although a few SNPs resulting in point muta-tions have been also reported[3,51,52]. However, Saison et al[53]. reported

that two Lan – mutations did not alter cell surface expression of ABCB6 in a heterologous expression system, suggesting the point mutation alone could not account for the Lan – phenotype. Like DUH, the Lan – genotype has variable clinical significance: hemolytic transfusion reactions vary from severe to none at all in Lan – individuals and hemolytic disease of the fetus and newborn range from mild to non-existent symptoms[54]. It is possible that compensation for ABCB6 function in Lan – individual varies based on the Lan – mutation itself, however no studies correlating Lan – mutation to phenotype have yet been reported.

It has been previously established that in response to phenylhy-drazine, a reagent that causes hemolytic anemia, *Abcb6* KO mice showed upregulation of heme biosynthesis and iron-related pathways to compensate, although not fully, for the loss in *Abcb6*[14]. Data from the SHIELD database showed that other heme biosynthesis genes are also highly expressed in the cochlea (Supplementary Fig. 13)[45], how-ever the RNA-Seq data from the *Abcb6* KO cochlea did not show upregulation of heme synthesis genes (Supplementary Fig. 14). Addi-tionally, data from the SHIELD database showed that other mito-chondrial ABC transporters involved in heme and iron homeostasis are highly expressed in the cochlea compared to other ABCB proteins (Supplementary Fig. 15), yet increased expression of these genes in the cochlea via RNA-Seq was not observed. The cochlea used for RNA-Seq were isolated from mice 2–3 months of age, when hearing is fully mature and before any age-related hearing loss. By contrast, the SHIELD data was collected from mice at embryonic day 16 and post-natal days 0, 4, 7, and 16[45], suggesting heme-synthesis pathways and ABCB proteins may be most important in developmental stages before hearing onset, which occurs around P10.

Yet, the possibility of compensation does not explain how DUH individuals display more of a phenotype than their *ABCB6*-null coun-terparts. These data may suggest that it is not simply the absence of ABCB6 that is problematic, but a combination of loss of ABCB6 and the presence of either misfolded or mis-localized ABCB6 that causes abnormalities. For the purpose of this study, we have assumed that ABCB6 monomers dimerize only with other ABCB6 monomers, and no other ABC transporters. Although RNA-Seq from the cochlea of WT and *Abcb6* KO mice showed significant changes in several hearing-related genes (Supplementary Fig. 14a–c), no significant differences in expression of B-subfamily ABC transporters were observed (Supple-mentary Fig. 14d). Analysis of the SHIELD database has shown that only a few ABC transporters are highly expressed in the inner ear, further limiting the possible heterodimer partners for ABCB6. Hetero-dimerization between ABCB6 and ABCB5 has been proposed[55], how-ever ABCB5 is not highly expressed in the cochlea according to the SHIELD database (Supplementary Fig. 15) and no significant expression change was identified by RNA-Seq between WT and *Abcb6* KO mouse cochleae. Therefore, we think it unlikely that heterodimerization

between Abcb6 and another B-subfamily ABC transporter is responsible for the inner ear phenotype observed in mice.

ABCB6 is a ubiquitously expressed transporter and is a N-glycosylated protein in some human tissues[56]. N-linked glycosylation of ABCB6 allows for differential trafficking through the ER and Golgi[41]. However, ABCB6 homologs from other species, including zebrafish, lack this glycosylation site (Fig. 3a). Zebrafish Abcb6 was determined to not be a glycoprotein (Fig. 3b) and did not localize to the plasma membrane (Supplementary Fig. 6e–g). Despite this difference, zebrafish Abcb6 bound ATP and porphyrins comparably to their human counterpart (Fig. 3c, d, respectively). The zebrafish Abcb6 homology model was obtained using the human ABCB6 cryo-EM structure as a template (Fig. 3e). Introduction of the L356P mutation in the zebrafish Abcb6 homology model also caused premature termination of the coupling helix adjacent to the L356P mutation (Fig. 3f). Although MD simulations suggested zebrafish Abcb6 may be more rigid than its human counterpart, taken together our results suggested human ABCB6 and zebrafish Abcb6 behave similarly, allowing us to move forward to study the function of Abcb6 in vivo.

In vertebrates, mechanosensory hair cells detect acoustic and vibrational stimuli for hearing and vestibular function. Our zebrafish studies show a link between Abcb6 and the anatomical structures responsible for mechanosensation associated with hearing and balance. WISH studies showed that zebrafish Abcb6 was prominently expressed in the otic vesicle (Fig. 4a), suggesting a role in inner ear development. Zebrafish embryos injected with a splice-blocking zebrafish Abcb6 MO15 developed fewer inner ear and lateral line hair cells than their WT counterparts (Fig. 4b and Supplementary Fig. 10). Injection with zebrafish Abcb6 MO15 prevented the development of the utricular otolith, an organ associated with vestibular function in most fishes and in mammals (Fig. 5a)[57–59]. This phenotype could be partially rescued by the injection with human ABCB6 mRNA (Fig. 5c). Zebrafish Abcb6 morphants showed altered swimming behavior compared to their WT counterparts, further evidence of lateral line and/or vestibular dysfunction (Fig. 6). While uninjected zebrafish swam smoothly in the upper 2/3 of the tank (Fig. 6a, f), the zebrafish Abcb6 MO15 morphants swam erratically and spent a significant amount of time in the lower third of the tank (39.9% compared with 1.2% for uninjected) (Fig. 6b, f). Abcb6 MO15 morphants swam significantly farther in the vertical direction (160.7 pixels compared with 79.7 pixels for uninjected), suggesting abnormalities in vestibular function (Fig. 6h). Some zebrafish Abcb6 MO15 morphant larvae had very few neuromasts at 3 dpf and did not show normal swim bladder inflation at 4 dpf, suggesting an overall developmental impairment. By 5 dpf, however, fish appeared morphologically normal with the otolith phenotype persisting for up to 14 dpf, suggesting a specific role for zebrafish Abcb6 in lateral line and inner ear development.

Both MO15- and MO7-injected morphants showed statistically significant reduction in hair cell numbers compared to their uninjected and control MO counterparts (Supplementary Fig. 10, 11). This reduction was observed using two different hair cell labels in multiple zebrafish strains. Although both MO15 and MO7 treatment significantly reduced hair cells, only MO15 produced the unique single-otolith phenotype (Supplementary Fig. 12). This phenotype was dependent on MO15 dosage, but not MO7 dosage (Supplementary Fig. 12b, c). Although MO7 did not produce a single-otolith phenotype like MO15, we believe the MO15 phenotype is specific to Abcb6. The single-otolith phenotype was observed in AB, nacre, TL, p53(-/-), and Tg(Brn3c:GFP) cell lines, and our morphants did not exhibit any gross morphological differences from their uninjected, control MO, and scrambled MO counterparts, suggesting the otolith phenotype was not due to off-target effects. Additionally, other MOs that affect the lateral line have not been reported to affect the otoliths[60–62]. However, even though we believe the single-otolith phenotype is specific to Abcb6, the specificity does not explain the observed differences between the MO7 and MO15 morphant. However, the phenotypic difference between MO15 and MO7 morphants elegantly parallels the variable expressivity of DUH and differing symptom severity of Lan – individuals. As previously stated, findings in the literature suggest that misfolded or mislocalized ABCB6 is more problematic than the absence of ABCB6. We suspect that the MO15, which targets exon15/intron15 splice junction, may allow for expression of a misfolded protein product that interferes with WT zebrafish Abcb6 function like a dominant negative allele. MO7, which targeted the exon7 splice site, likely does not prevent WT Abcb6 function. Future studies are necessary to further explain the discrepancy between MO7 and MO15 treatment as well as how the hair cells and otolith are affected. o-Dianisidine staining in zebrafish showed that red blood cells were not affected by MO15 treatment (Supplementary Fig. 16), suggesting a compensation mechanism for at least the role of zebrafish abcb6 in heme biosynthesis.

Data found in the Shared Harvard Inner-Ear Laboratory (SHIELD) database showed mouse Abcb6 mRNA counts increased with development in the cochlea, the portion of the mammalian inner ear required for hearing (Fig. 7a)[45]. Conversely, mRNA counts in the utricle, which is important for balance, did not show a linear increase with development. Instead, there was a marked increase in ABCB6 expression in the utricle between E16 and P0, consistent with the developmental timing of hair cell mechanosensory function in this epithelium[63]. Our preliminary physiology studies also suggest a role for ABCB6 in mammalian auditory function. Using auditory brainstem response (ABR) recordings, a gross physiological measure of hearing sensitivity, we show that Abcb6 knockout mice have increased ABR threshold at all tested frequencies, as compared to the WT mouse, indicating reduced hearing sensitivity (Fig. 7b). These data are consistent with our zebrafish data demonstrating that Abcb6 plays an important role in the inner ear. However, the mammalian cochlea is morphologically more complex than the zebrafish inner ear, with multiple subtypes of hair cells and supporting cells. Future studies will be needed to determine how ABCB6 mutations and knockdown impact cochlear development in mammalian models. Determining the localization of ABCB6 in the inner ear will help direct our future studies in mouse models as well.

ABCB6 is not the first ABC transporter linked to the inner ear: mutations in ABCC1 and ABCA12 have been linked to hearing abnormalities[64,65]. Several ABCB6 mutations have been found in DUH patients, yet it is currently unknown if specific mutations are associated with more severe cases of DUH. Further studies are required to determine if certain DUH mutations present with a greater frequency of hearing loss. Family members with the same ABCB6 mutation may exhibit DUH with differing severity, suggesting a variable penetrance of DUH[66], as observed for Lan – individuals[52]. Additional investigation of how ABCB6 phenotype is modulated may shed light on the heterologous nature of this orphan disease and other ABCB6-related diseases.

## Methods

### Ethics

All experiments were performed in compliance with all relevant ethical regulations in accordance with the University of Wisconsin-Madison Institutional Animal Care and Use Committee (IACUC) (protocol #M005020), Washington State University IACUC (protocol #6024), the Animal Care and Use Committee at the National Institutes of Health (NIH; protocol #1362-13), or by the St. Jude IACUC (animal protocol #297).

### Reagents

Dulbecco's Modified Eagle Medium (DMEM), 10% FBS, L-glutamine, and phosphate-buffered saline (PBS) were purchased from Gibco. DMEM was also purchased from Lonza. Lipofectamine 2000, Lipofectamine Plus, Opti-MEM reduced serum media, and NuPAGE LDS Sample buffer were purchased from Thermo Fisher Scientific. Protease

inhibitor tablets (cOmplete™ EDTA-free protease inhibitor cocktail) were purchased from Roche. Hemin-Agarose beads (Hemin-Agarose Type I, saline suspension), ATP-agarose beads (Adenosine 5′-Triphosphate-Agarose), and Flag M2® resin were purchased from Millipore Sigma. Detergents and cholesteryl hemisuccinate were purchased from Anatrace. Peptide-N-glycosidase F, restriction enzymes, and cloning reagents were purchased from New England Biolabs. Porphyrins and other chemicals were purchased from Millipore Sigma and Frontier Scientific. DNA primers were purchased from Invitrogen. Antibodies were purchased from Rockland Imunochemicals (Anti-ABCB6, 600-401-945, 1:1000 dilution), Proteintech (Anti-GAPDH, 60004-1-Ig, 1:10,000 dilution), Jackson ImmunoResearch Laboratories Inc (Peroxidase AffiniPure Donkey Anti-Mouse IgG (H + L), 715-035-150, 1:10,000; Peroxidase AffiniPure Donkey Anti-Rabbit IgG (H + L), 715-035-152, 1:10,000 dilution), Cell Signaling Technology (Anti-Phospho-eIF2α (Ser51) Antibody #9721, 1:1000 dilution), Santa Cruz Biotechnology (GFP Antibody (B-2) HRP, sc-9996 HRP, 1:1000 dilution), and Genescript (custom-developed Anti-ABCB6, 1:1000 dilution).

### Sequencing alignments
Sequences of ABCB6 from *Homo sapiens* (human, UniProt Q9NP58), *Pan troglodytes* (chimpanzee, UniProt A0A6D2WHF9), *Mus musculus* (mouse, UniProt Q9DC29), and *Danio rerio* (zebrafish, UniProt A0A2R8RXI2) were aligned using Clustal Omega in Megalign Pro V.17.0.2. Pairwise alignments showed human ABCB6 showed 99.5% identity with the chimpanzee homolog, 89.0% identity with the mouse homolog, and 63.6% identity with the zebrafish homolog. Pairwise alignments were calculated using Semi-Global: Needleman-Wunsch alignment and a BLOSUM62 substitution matrix.

### Cell culture
HEK-293 cells were maintained in Lonza DMEM (4.5 g/L glucose) supplemented with 10% FBS, 2 mM L-glutamine, 100 units/mL penicillin, and 100 μg/mL streptomycin in humidified 5% $CO_2$ at 37 °C. NIH 3T3 cells were maintained in Lonza DMEM (4.5 g/L glucose) supplemented with 10% BCS, 1 mM sodium pyruvate, 1X MEM Non-essential Amino Acids (Gibco 11140050), 2 mM L-glutamine, 100 units/mL penicillin, and 100 μg/mL streptomycin in humidified 5% $CO_2$ at 37 °C.

### Cloning of DUH variants
ABCB6 variants of DUH mutations were cloned using a QuikChange II XL site-directed mutagenesis kit on a previously constructed pCDNA-3.1-ABCB6-flag plasmid[11] as a template. PCR primers are included in Supplementary Table 5.

### Small-scale expression of ABCB6 and mutants
HEK 293 cells were transfected with pcDNA-3.1-ABCB6-flag plasmid using lipofectamine 2000 according to manufacturer's instructions. Twenty-four hours after transfection, cells were harvested, rinsed with PBS, and resuspended in lysis buffer (M-PER® Mammalian Protein Extraction Reagent (proprietary detergent reagent) supplemented with Roche cOmplete™ EDTA-free protease inhibitor cocktail, 1 mM phenylmethylsulfonyl fluoride (PMSF), 1 μM MG132, and 10 mM N-Ethylmaleimide (NEM)). Resuspended pellets were solubilized by vortexing briefly(<1 s), then incubating on ice for 30 min with brief vortexing every 10 min. Lysates were centrifuged at 15,000 x g in a pre-chilled centrifuge for 10 min to pellet insoluble debris. The supernatant was transferred to a fresh tube and protein content was quantitated using the Pierce BCA Protein Assay. 15–30 μg of lysate was loaded in each lane of a gel (μg of lysate loaded was consistent for each ABCB6 mutant lysate across each gel), bands were transferred to a nitrocellulose membrane, and the western blot was developed using one of the Anti-ABCB6 antibodies described previously. Replicates were performed by repeating the experiment in its entirety (transfection to western blot).

### ATP-agarose pulldown assays
ABCB6 mutants were expressed and cell lysates were prepared as described above. 100 μL of dry ATP beads were measured into a 1.5 mL Eppendorf tube. 900 μL of sterile MilliQ water was added to the tube and the contents were mixed by pipette. The tube was incubated at ambient temperature for 30 min on a tube rotator. The beads were centrifuged at 1500 x g for 5 min and the supernatant was aspirated off. The beads were washed an additional 3 times with 900 μL of sterile MilliQ water. After the last wash, beads were resuspended to give a 50% v/v slurry in lysis buffer. 50 μL of the slurry was transferred to each 1.5 mL Eppendorf tube and the beads were pelleted at 1500 x g for 5 min. After pelleting, the supernatant was removed and 200 μg of each clarified cell lysate (0.7 mg/mL protein concentration) was added to its respective tube of beads. Clarified lysates were incubated on the beads overnight at 4 °C on a tube rotator with vigorous mixing. After incubation, beads were pelleted as previously described and supernatants were transferred to fresh tubes. The beads were washed three times with 500 μL lysis buffer with 5-minute incubation on the tube rotator before centrifugation. Beads were resuspended in 50 μL of elution buffer (1X NuPAGE LDS Sample Buffer, 5% 2-mercaptoethanol, 8 M urea, and MPER buffer) and heated at 37 °C for 30 min with 1000 RPM shaking to elute sample off the beads. Samples were centrifuged at 2000 x g for 5 min and the eluted supernatant was removed from the beads. Eluted samples were analyzed by western blot using an Anti-ABCB6 antibody. Signal of the input and eluted bands were quantitated using ImageStudio Software V5.2.5. Formulas for the percent of WT binding can be found in the Supplementary Methods. Replicates were performed by repeating the experiment in its entirety (transfection to western blot).

### Hemin-agarose pulldown assays
ABCB6 mutants were expressed as described above and cell lysates were prepared in lysis buffer. 200 μg of clarified lysate was diluted with lysis buffer for a 0.35 mg/mL final protein concentration. Hemin-agarose beads were centrifuged at 4500 x g for 5 min and the supernatant was aspirated off. The beads were resuspended to 50% v/v in lysis buffer, mixed by pipette, and then centrifuged as previously described. The supernatant was removed, and the beads were washed again with 500 μL lysis buffer for a total of 3 times. Beads were resuspended as a 50% v/v slurry in lysis buffer and 200 μL of the slurry was added to each amber 1.5 mL Eppendorf tube. The beads were pelleted, the supernatant was aspirated, and each lysate mixture was added to its own tube of hemin-agarose beads. The lysate-bead mixture was left to incubate on the hemin-agarose beads overnight at 4 °C on a tube rotator with vigorous mixing. The beads were then pelleted as previously described as the supernatant was removed. The beads were washed three times with 500 μL lysis buffer with 5 min incubation on the tube rotator before centrifugation. The beads were then pelleted as previously described as the supernatant was removed. Beads were resuspended in 50 μL of elution buffer and heated at 37 °C for 30 min with 1000 RPM shaking to elute sample off the beads. Samples were centrifuged at 4500 x g for 5 min and the eluted supernatant was removed from the beads. Eluted samples were analyzed by western blot using an Anti-ABCB6 antibody. Signal of the input and eluted bands were quantitated using ImageStudio Software V5.2.5. Formulas for the percent of WT binding can be found in the Supplementary Methods. Replicates were performed by repeating the experiment in its entirety (transfection to western blot).

### Treatment of L356P with proteasomal and lysosomal inhibitors
HEK 293 cells were seeded into 6-well plates and left overnight. The following day, cells were transfected with each DUH variant using lipofectamine 2000 according to manufacturer's instructions. Supplementary Table 6 outlines treatment conditions for each compound. All compounds were prepared in PBS except for MG132, which was prepared in DMSO then diluted in PBS. After adding compound, cells

were returned to the incubator. After treatment was complete, cells were harvested, lysed, and quantitated as previously described. Band signals were quantitated using ImageStudio Software V5.2.5. Calculations for the percent of untreated signal can be found in the Supplementary Methods. Replicates were performed by repeating the experiment in its entirety (transfection to western blot).

## Large-scale expression and purification of ABCB6

ABCB6 was expressed in Expi293F cells (ThermoFisher Scientific, A14527) by transient transfection using FectoPro according to manufacturer protocol (pCDNA vector) or using baculo virus (pEG vector). ABCB6 expression was boosted with sodium butyrate or 4-PBA (L356P) after 24 h. After 48 h, cells were harvested by centrifugation using a JA-14 fixed-angle rotor at 6000 RPM (5520 x g). Cell pellets were resuspended in 10 mM Tris, pH 7.5, 150 mM NaCl, 1% w/v n-dodecyl-β-maltoside (DDM), 0.2% w/v cholesteryl hemisuccinate (CHS) and solubilized for 1 hour at 4 °C on a tube rotator. Solubilized lysate was pelleted using a JA-25.5 fixed-angle rotor at 24,000 RPM (69,970 x g) for 30 min at 4 °C and the supernatant was applied to Flag M2® Affinity Gel. The resin was washed with 20 column volumes of wash buffer (10 mM Tris, pH 7.5, 150 mM NaCl, 0.1% w/v DDM, 0.02% w/v CHS). ABCB6 was eluted in wash buffer supplemented with 200 µg/mL of 3X Flag peptide, incubating each fraction for 5 min. Eluted fractions were pooled, concentrated, and purified by size exclusion on a Superose 6 column equilibrated with 10 mM Tris, pH 7.5, 150 mM NaCl, 0.01% w/v DDM, 0.002% w/v CHS.

## Cryo-EM data collection

The cryo-EM grids were prepared using Vitrobot Mark IV (FEI) operated at 4 °C and 100% humidity. For samples of ABCB6 WT, 3 µL aliquots of samples at concentrations of approximately 3.6 mg/mL were applied onto glow-discharged holey carbon grids (Quantifoil R1.2/1.3) 400-mesh Au grid. After a waiting time of 10 s, the grids were blotted for 2 s and plunged into liquid ethane for quick freezing. The cryo-EM grids were screened on a Titan Krios microscope (FEI) operated at 300 kV using a Gatan K2 Summit detector equipped with a GIF Quantum energy filter. Images were automatically recorded using SerialEM[67] with a slit width of 20 eV on the energy filter and in super-resolution mode at a nominal magnification of 130,000×, corresponding to a calibrated pixel size of 0.5436 Å at object scale, and with defocus ranging from 0.5 to 2.5 µm. Each stack was exposed for 3 s with an exposing time of 0.05 s per frame, resulting in a total of 60 frames per stack, and the total dose rate for each stack was about 66 e⁻/Å2.

## Cryo-EM data processing

Motion correction was performed in cryoSPARC[68] prior to binning two-fold to 1.0872 Å/pixel. Patch contrast transfer functions (CTF) was estimated for each micrograph. The micrographs were manually curated for ice quality and thickness and CTF fit. 3,344,054 particles were initially picked from 7013 micrographs. After a single round of 2D classification, the selected classes accounted for 2,846,893 particles. Three ab initio classes were generated from the selected particles comprising a ABCB6 class, a micelle class, and a junk class. After a single round of hetero refinement, particles from the ABCB6 class were aligned using non-uniform refinement with no symmetry applied. 3D classification without alignment was unsuccessful in improving the resolution and quality of the reconstructions. Particles were aligned following global and local CTF refinement using non-uniform refinement with a mask omitting the detergent micelle and with C2 symmetry applied.

## Model building, refinement, and validation

The coordinates from PDB ID: 7DNY[17] were docked into the reconstruction. One of the half maps was used for real-space refinement in PHENIX[69]. A sharpened map was generated using DeepEMhancer[70].

The final model was validated using MolProbity[71]. Figures were created in ChimeraX[72] and PyMOL[73,74].

## ATPase assays

For each sample, 5 µg of purified protein in (10 mM Tris, pH 7.5, 150 mM NaCl, 0.01% w/v DDM, 0.002% w/v CHS) was suspended in 50 µL total volume reaction buffer (40 mM MOPS (pH 7.0), 10 mM MgCl₂, 50 mM KCl, 2 mM DTT, 0.5 mM EDTA, 5 mM sodium azide, and 1 mM ouabain). ATP solution was freshly prepared in reaction buffer at double the intended final concentration and brought to pH 7.0. To start the reactions, 50 µL of ATP solution was added to each sample, mixed briefly, then incubated for 30 min at 37 °C. Blanks with identical conditions, but lacking magnesium, were prepared for each sample condition. Reactions were stopped by immediate immersion in ice and directly centrifuged at 21,000 x g in a refrigerated centrifuge. 50 µL of the supernatant was transferred to a 96-well plate and reacted with and equal volume of developing mix, a 1:1 mixture, made immediately before use, of 1% ammonium molybdate in water and freshly prepared 6% ascorbate in 1 N HCl. Plates were allowed to stand for 2 min for full color to develop and read at a wavelength of 800 nm using a BioTEK Synergy H4 Plate Reader. Quantitation was performed using phosphate standards diluted in reaction buffer in a range of 0–50 nmols, with standards and samples being treated identically. $N = 15$ experiments for ABCB6 WT from five biological replicates, and $N = 8$ experiments for E752Q from three biological replicates, $N = 4$ for L356P from two biological replicates.

## Isothermal shift assays

The isothermal shift assay was adapted according to Ashok et al.[75]. Briefly, purified proteins (0.25 µg ABCB6 WT protein or 0.5 µg ABCB6-L356P protein/20 µL final reaction volume) in 1x assay buffer (SB PREDIVEZ™ Reagent Kit for BCRP, SBPVR4, Solvo Biotechnology) were heated in a thermocycler for 3 min at various temperatures (37–75 °C) to establish a thermal denaturation curve. Samples were then treated with ice-cold PBS supplemented with NP-40 to a final concentration of 0.8%. Subsequently, ultracentrifugation (at 100,000 x g for 20 min at 4 °C) was performed to precipitate the denatured proteins. The supernatant (20 µL) was subjected to immunoblot analysis using an ABCB6 antibody (Genescript custom-developed Anti-ABCB6, rabbit 2° antibody). Based on an extrapolation from the thermal denaturation curve that produced 99% loss of protein in the supernatant, 51 °C was selected for ABCB6 WT and 65 °C for L356P (Supplementary Fig. 1f). To assess the ability of AMP-PNP to thermally stabilize ABCB6, proteins were incubated with AMP-PNP for 60 min at 37 °C. Samples were then heated to the previously selected temperatures for 3 min. The signal intensity of the heated sample was normalized to the signal intensity of the unheated samples (ABCB6 WT and L356P) and the signal intensity reported as percent (%) of unheated control. Three technical replicates were performed on the same batch of protein used for ATPase assays.

## Molecular dynamics simulations of WT and L356P

The structure of ABCB6 WT was taken from the determined cryo-EM structure (EMDB ID: EMD-46724, PDB ID: 9DBQ). The L356P mutant variant was created by introducing the L356P mutation into ABCB6 WT using PyMOL[73,74]. The PACKMOL-memgen module of AmberTools20 was used to embed the transporters in a phospholipid membrane, solvate, add ions and perform the initial minimization of the system[76,77]. A model membrane consisting of 23% CHOL/29% OAPE/48% PSPC in the extracellular leaflet and 26% CHOL/50% OAPE/24% PSPC in the intracellular leaflet was used for all simulations to reflect the composition of an erythrocyte plasma membrane[78,79]. Each system was solvated with TIP3P water and neutralized with 0.15 M NaCl. Supplementary Table 8 outlines the composition of each of the systems simulated. Simulations were performed in the apo state, in the absence of ATP and transport substrate. All systems were simulated using

Amber20 in conjunction with the Amber ff19SB force field[80] for the protein, Lipids 21 force field[81] for the lipids and monovalent ion parameters from Joung & Cheatham[82]. Systems were energy minimized and equilibrated using the default parameters from PACKMOL-memgen[77]. Briefly, the systems were energy minimized using the 5000 steps of steepest descent minimization, followed by 5000 steps of conjugate gradient minimization. Subsequently, over 1 ns, systems were heated to 310 K with a 10.0 kcal/mol Å$^2$ restraint on the protein, and then equilibrated without restraints for 5 ns. Each system was then simulated in triplicate for 500 ns without restraints. In all simulations, a 2 fs timestep and SHAKE on bonds with hydrogen were used. The pressure was maintained at 1 bar using an anisotropic Berendsen barostat and the temperature was maintained at 310 K using the Langevin thermostat. Periodic boundary conditions were implemented. Analysis was performed on the full 500 ns unrestrained simulation using AmberTools20[76] and Python 3. Equilibration of each system was confirmed via plateauing of the time-dependent backbone RMSD (Supplementary Fig. 17). The Visual Molecular Dynamics (VMD) program[83] and PyMOL[73,74] were used for visualization of the simulations.

### Cloning of zebrafish Abcb6 by RT-PCR

Embryos at 5 days post-fertilization were anesthetized in 0.02% Tricaine and transferred into RNase/DNase-free 1.5 ml microcentrifuge tubes with fitted pestle (Kontes). Approximately 50 embryos were homogenized in 200 µL TRIzol and total RNA was extracted according to the manufacturer's protocol (Invitrogen). cDNA was synthesized by reverse transcription using the SuperScript III First-Strand Synthesis System with Oligo(dT) primers according to the manufacturer's protocol (Invitrogen). Full length zebrafish abcb6 was PCR amplified from cDNA using AccuPrime Taq DNA Polymerase (Invitrogen) with the following DNA primers: abcb6 Forward 5′-CTTCATCATGGTGGA-GATGAAGAG-3′ and abcb6 Reverse 5′-CGTGTCCCCTGTGTGCTGTAG-3′. The subsequent 2561 bp product was purified using the QIAquick PCR Purification Kit (Qiagen), then cloned into the multiple cloning site of pcDNA3.1 (Thermo Fisher) containing a C-terminal flag tag using the HindIII and EcoRI cut sites. The gene was also cloned into a pcDNA3.1 vector containing a C-terminal GFP tag.

### Peptide-N-glycosidase F (PNGase F) assay

PNGase assays were performed as described previously[41]. Briefly, NIH3T3 cells were transiently transfected with pCDNA-3.1 expression plasmids outlined above using Lipofectamine Plus according to manufacturer's protocols. Twenty-four hours after transfection, cells were washed with PBS, scraped into 1 mL of cold PBS containing 1X cOmplete™ EDTA-free protease inhibitor cocktail. Cells were pelleted by centrifugation at 1000 x $g$ for 4 min at 4 °C and solubilized in buffer A (50 mM Tris-HCl, pH 7.5, 150 mM NaCl, 10% glycerol, 1% Nonidet P40, and 1X cOmplete™ protease inhibitor cocktail). Lysates were denatured in 1X denaturing buffer (0.5% SDS and 1% β-mercaptoethanol) and incubated at 37 °C for 1 hour in reaction buffer (50 mM Na$_2$PO$_4$, pH 7.5 and 1% Nonidet P-40) with or without PNGase F according to manufacturer's recommendations.

### ATP-agarose pulldown assays with zebrafish Abcb6

ATP-agarose pulldowns were performed as described for the DUH mutants except using either a pcDNA-3.1 zebrafish Abcb6-Flag or -GFP expression constructs.

### Hemin-agarose pulldown assays with porphyrin competition

Flag-tagged human ABCB6 and zebrafish Abcb6 were expressed as described above (small scale expression) and cell lysates were prepared in lysis buffer. 200 µg of clarified lysate (0.35 mg/mL final protein concentration) and porphyrin (50 µM final concentration) were mixed and incubated for 1 hour at ambient temperature with gentle rocking. Hemin-agarose beads were centrifuged at 4500 x $g$ for 5 min

and the supernatant was aspirated off. The beads were resuspended to 50% v/v in lysis buffer, mixed by pipette, and then centrifuged as previously described. The supernatant was removed, and the beads were washed again with 500 µL lysis buffer for a total of 3 times. Beads were resuspended as a 50% v/v slurry in lysis buffer and 200 µL of the slurry was added to each amber 1.5 mL Eppendorf tube. The beads were pelleted, the supernatant was aspirated off, and each lysate mixture was added to its own amber tube of hemin-agarose beads. The lysate-bead mixture was left to incubate on the hemin-agarose beads overnight at 4 °C on a tube rotator with vigorous mixing. The beads were then pelleted as previously described and the supernatant was removed. The beads were washed three times with 500 µL lysis buffer with 5 min incubation on the tube rotator before centrifugation. The beads were then pelleted as previously described as the supernatant was removed. Beads were resuspended in 50 µL of elution buffer and heated at 37 °C for 30 min with 1000 RPM shaking to elute sample off the beads. Samples were centrifuged at 4500 x $g$ for 5 min and the eluted supernatant was removed from the beads. Eluted samples were analyzed by western blot using an Anti-ABCB6 antibody. Signal of the input and eluted bands were quantitated using ImageStudio Software V5.2.5. Formulas for the percent of WT binding can be found in the Supplementary Methods. Replicates were performed by repeating the experiment in its entirety (transfection to western blot).

### Generation of zebrafish homology model

Homology modelling of zebrafish ABCB6 was conducted by employing UCSF Modeller 9.23[84]. The atomic coordinates of the human ABCB6 obtained in this study (64% of sequence identity) were employed as 3D template. The TMD$_0$ of ABCB6 was not modeled. Among different eukaryotic species, the sequence corresponding to the internal loop area, which defines the limits of the binding cavity, is almost entirely conserved[18]. Therefore, rather than using a bacterial Atm1/ABCB7/HMT1/ABCB6 ortholog template where the internal loop area is absent, a eukaryotic template was used[85]. Given the conformational flexibility of the internal loops, 250 homology models were initially generated, and DOPE and molpdf were employed for model ranking. The final model was chosen based on a consensus of the top ranked results from both scores. Further investigations with different approaches were also conducted, however these did not lead to any substantial differences with respect to the applied methodology. The final homology model was compared to the structure obtained through AlphaFold[39] with a good agreement in terms of RMSD, except for the coupling helices.

### Molecular dynamics simulations of zebrafish homology model

Molecular dynamics simulations on the zebrafish homology model were performed as described for the human ABCB6 model above.

### Zebrafish

AB, nacre, p53 (-/-), and *Tg*(Brn3c:GFP), and TL zebrafish strains were used. Zebrafish were maintained and bred using standard practices[86]. Embryos and larvae were maintained at 28.5 °C in egg water (0.03% Instant Ocean in reverse osmosis water). All experiments were performed in accordance with the University of Wisconsin-Madison IACUC (protocol #M005020), Washington State University IACUC (protocol #6024), or by the Animal Care and Use Committee at the National Institutes of Health (NIH; protocol #1362-13).

### Whole-mount in situ hybridization (WISH)

WISH was performed as previously described[87]. Briefly, larvae were fixed in 4% paraformaldehyde at 4 °C overnight, dehydrated in 100% methanol and stored at −20 °C. Larvae were rehydrated in PBS, permeabilized with proteinase K, prehybridized at 70 °C for 2 h and hybridized with digoxigenin (DIG)-labeled RNA probes at 70 °C overnight. For probe synthesis, *abcb6a* was PCR amplified from wild-type

zebrafish cDNA and cloned into the pCRII-TOPO Dual Promoter Vector (Invitrogen). Sense and antisense DIG-labeled RNA probes were synthesized with Sp6 and T7 RNA polymerase using a DIG RNA Labeling Kit (Roche). After washing and blocking, the larvae were incubated with anti-DIG-AP Fab fragment for 2 h and stained with BCIP/NBT substrate until the desired signal intensity appeared. Stained larvae images were captured using a Nikon SMZ1500 stereomicroscope equipped with a Nikon DS-Fi2 color camera and Nikon NIS-Elements software.

## Zebrafish Abcb6 morpholino design and microinjection

Morpholino antisense oligonucleotides (MO) were designed by GeneTools, LLC (Philomath, OR) to target two independent splice donor sites in the zebrafish *abcb6a* gene (). The following MO sequences were used: *abcb6a* exon7/intron7 (MO7) 5′-ACCATTGTTAAA-TACTCACTGAGGT-3′ and *abcb6a* exon15/intron15 (MO15) 5′-GAT-TATTGTCAGATTCACCTTTGAG-3′ (splice donor sites are unlined). The sequence of MO15 was scrambled to create the Scrambled MO 5′-ATGTAGTTCTTATAGGCTTACGTCA- 3′. The control MO sequence was 5′-CCTCTTACCTCAGTTACAATTTATA-3′, which is a standard control available from GeneTools, LLC (PCO-StandardControl-100). MOs were resuspended at a stock concentration of 1 or 2 mM in dH$_2$O. For microinjection, MOs were diluted to 0.2, 0.5, and 1.0 mM in dH$_2$O containing 0.05% phenol red as an injection tracer. Microinjection needles were fabricated from 1.2 mm thin wall glass capillaries (WPI; TW120F-4) using a Sutter Instrument Flaming/Brown Micropipette Puller (Model P-87). The pulled needles were top loaded with MO solution, the needle tips were clipped open to 10–20 μm using fine forceps, and the injection volume was calibrated by injecting into mineral oil and measuring the droplet diameter using a 0.01 mm Stage Micrometer (Fisher Scientific, NC9167561). Microinjections were performed using a Pneumatic PicoPump (WPI, PV820) combined with a Nikon stereomicroscope (Nikon, SMZ645) and a manual micromanipulator (WPI, M3301). Freshly fertilized single-cell embryos were aligned in an 2% agarose mold (WPI, Z-MOLDS) and microinjected into the yolk with 1 nL of MO solution to deliver 8, 4, 2, 1, or 0.5 ng of MO. Embryos damaged during microinjection were removed from further analysis. No obvious off-target effects were observed with any of the *abcb6a* MOs. Control and *abcb6a* morphants at 1–5 dpf were imaged (then euthanized and fixed in 4% PFA), or collected for RNA analysis to validate gene knockdown. Knockdown effectiveness was determined by RT-PCR (Supplementary Methods).

## Zebrafish hair cell studies

This experiment used *Tg*(Brn3c:GFP) transgenic fish, which express membrane-bound green fluorescent protein (GFP) in lateral line and inner ear hair cells[88]. Zebrafish embryos were generated by paired matings of adults reared in the Coffin Lab facility at Washington State University Vancouver. Morpholinos were injected into newly fertilized eggs generated by natural mating. Fish were reared in Petri dishes kept at 28 °C in fish water until 3 or 4 dpf, then euthanized by an overdose of MS-222, fixed in 4% paraformaldehyde, and labeled with anti-GFP using our published methods[60]. Fish were viewed with a Leica DMRB epifluorescent microscope using a 40X air objective (400X total magnification). For the lateral line we quantified hair cells in the same 7 neuromasts per fish (3 head, 2 trunk, 2 tail) For the inner ear, we quantified all GFP+ hair cells in the anterior macula (associated with the utricular otolith). Data were analyzed by 1- or 2-way ANOVA (for inner ear or lateral line, respectively) using GraphPad Prism v9. Experiments were repeated at least once.

## Zebrafish behavioral analysis

Video files of zebrafish (Supplementary Movie 1 and 2) were analyzed using the MTrackJ plugin for Image J. Tracking data was exported using the "Measure" function and further analyzed by an unpaired two-tailed Welch's T-test using GraphPad Prism 8.

## *Abcb6* KO mice

Animal studies were conducted following the protocols approved by Institutional Animal Care and Use Committee at St. Jude Children's Research Hospital (animal protocol #297). All mice were born and housed within an AAALAC-accredited animal facility with 24/7 veterinary care. Mice were maintained on a standard rodent diet with food and water *ad libitum*. The *Abcb6* KO mice were previously generated in our lab[14]. Mixed 129/ C57BL/6 background mice were used for auditory brainstem response testing outlined below. A mixed background was chosen due to the well-known age-related hearing loss of C57BL/6 mice.

## Auditory Brainstem Response (ABR) testing

ABR waveforms were recorded in a sound booth (Industrial Acoustic Company) by using subdermal needles positioned in the skull, below the pinna, and at the base of the tail, and the responses were fed into a low-impedance Medusa digital biological amplifier system (RA4L; TDT; 20 dB gain). At each frequency, the stimulus intensity was reduced from 90 to 0 dB in 5-dB steps to determine the threshold decibel sound pressure level (SPL) when the electrical response was just above the noise floor. ABR waveforms were averaged in response to 500 tone bursts. The recorded signals were filtered by a band-pass filter from 300 Hz to 3 kHz. Individual ABR wave 1 amplitudes were measured as the difference between the positive peak and the following negative trough.

## Statistics and reproducibility

Sample sizes were chosen based on standards in the field, with $N = 5$ or 6 mice and $N > 8$ fish for zebrafish studies. Statistical analyses were performed using GraphPad Prism using an unpaired two-tailed Welch's T-test, 1- or 2-way ANOVA as described in the methods, figure captions, or source data. The only data excluded from the study were two datapoints from the ATPase dataset identified by GraphPad Prism using a ROUT method with Q = 0.1% to remove only definite outliers. Zebrafish embryos damaged during microinjection were removed from further analysis. All experiments were repeated at least once and were successful. For experiments using transient transfections (small scale expressions of DUH, ATP- and hemin-agarose pulldowns with DUH mutants or zebrafish Abcb6), each biological replicate was repeated from transfection to western blot. Due to the low expression of L356P and the tendency to misfold, assays using purified L356P were limited. For the ATPase assays, $N = 15$ experiments for ABCB6 WT from five biological replicates, and $N = 8$ experiments for E752Q from three biological replicates, $N = 4$ for L356P from two biological replicates. Thermal shift assays were performed using three technical replicates of ABCB6 WT and three technical replicates of L356P, using the same batches of protein as were used for the ATPase assays. Thermal stability studies were performed on three technical replicates of ABCB6 WT and two technical replicates of L356P, using the same batches of protein as were used for ATPase assays and thermal shift assays. Technical replicates were distinct samples performed on consecutive days. Peptide-N-glycosidase F (PNGase F) assay was performed only once as it was a qualitative assay meant to confirm the "N-X-C" glycosylation motif was not present.

For zebrafish studies, each experiment was repeated, and the results showed the same pattern. Exact hair cell numbers varied from animal to animal, even within a group. Fertilized eggs were randomly assigned to the control or MO group. Eggs were harvested every 15–25 min in the morning during spawning. 10–20 eggs were injected with morpholino (MO), then the next 10–20 eggs would remain uninjected (or injected with scrambled MO), then the next set was injected with abcb6 MO, and so on. In lab zebrafish, sex determination is partially based on environment and sex cannot be detected in animals this young. However, based on research in the Coffin lab we generate approximately equal sex ratios once the fish are older.

ABCB6 KO and WT control mice included roughly equal numbers of males and females: $n = 3$ females and $n = 3$ males for ABCB6 KO, $n = 3$ females and $n = 2$ males for WT control mice.

## Reporting summary

Further information on research design is available in the Nature Portfolio Reporting Summary linked to this article.

## Data availability

The cryoEM structure is available in the Electron Microscopy Data Bank (EMDB ID: EMD-46724) and the Protein Data Bank (PDB ID: 9DBQ). Simulation data is available at https://github.com/OMaraLab/ABCB6 (https://doi.org/10.5281/zenodo.13363754). Data generated in this study are provided in the Supplementary Information or in the Source Data File. Additional data utilized in this study are as follows: Protein Data-bank Structures 4AYT, 7DNY, 7DNZ, 7D7N, 7D7R, and 7EKM; UniProt Protein Sequences A0A6D2WHF9, A0A2R8RXI2, O70595, Q9DC29, and Q9NP58; NCBI Protein Sequences NP_001072643.1 [https://www.ncbi.nlm.nih.gov/protein/NP_001072643.1], NP_001091625.1 [https://www.ncbi.nlm.nih.gov/protein/NP_001091625.1], NP_001139165.1 [https://www.ncbi.nlm.nih.gov/protein/NP_001139165.1], XP_004467845.2 [https://www.ncbi.nlm.nih.gov/protein/XP_004467845.2?report=genpept], XP_006890304.1 [https://www.ncbi.nlm.nih.gov/protein/XP_006890304.1], XP_007520645.1 [https://www.ncbi.nlm.nih.gov/protein/XP_007520645.1], XP_015145572.2 [https://www.ncbi.nlm.nih.gov/protein/XP_015145572.2], XP_018091199.1 [https://www.ncbi.nlm.nih.gov/protein/XP_018091199.1], XP_036029275.1 [https://www.ncbi.nlm.nih.gov/protein/XP_036029275.1]; AlphaFold Model F1QCK2; Ensembl ENSDARG00000063297, [https://nov2020.archive.ensembl.org/Danio_rerio/Gene/Splice?db=core;g=ENSDARG00000063297;r=1:6135176-6158609]; and Shared Harvard Inner-Ear Laboratory Database (SHIELD) Abcb1a [https://shield.hms.harvard.edu/viewgene.html?gene=Abcb1a], Abcb1b [https://shield.hms.harvard.edu/viewgene.html?gene=Abcb1b], Abcb10 [https://shield.hms.harvard.edu/viewgene.html?gene=Abcb10], Abcb2 (Tap1) [https://shield.hms.harvard.edu/viewgene.html?gene=Tap1], Abcb3 (Tap2) [https://shield.hms.harvard.edu/viewgene.html?gene=Tap2], Abcb5 [https://shield.hms.harvard.edu/viewgene.html?gene=Abcb5], Abcb6 [https://shield.hms.harvard.edu/viewgene.html?gene=Abcb6], Abcb7 [https://shield.hms.harvard.edu/viewgene.html?gene=Abcb7], Abcb8 [https://shield.hms.harvard.edu/viewgene.html?gene=Abcb8], Abcb9 [https://shield.hms.harvard.edu/viewgene.html?gene=Abcb9], Alad [https://shield.hms.harvard.edu/viewgene.html?gene=Alad], Alas1 [https://shield.hms.harvard.edu/viewgene.html?gene=Alas1], Cpox [https://shield.hms.harvard.edu/viewgene.html?gene=Cpox], Fech [https://shield.hms.harvard.edu/viewgene.html?gene=Fech], Hmbs [https://shield.hms.harvard.edu/viewgene.html?gene=Hmbs], mitoferrin (Slc25A37) [https://shield.hms.harvard.edu/viewgene.html?gene=Slc25a37], Ppox [https://shield.hms.harvard.edu/viewgene.html?gene=Ppox], Urod [https://shield.hms.harvard.edu/viewgene.html?gene=Urod], and Uros [https://shield.hms.harvard.edu/viewgene.html?gene=Uros]. Source data are provided with this paper.

## Code availability

Code and simulation data is available at https://github.com/OMaraLab/ABCB6 (https://doi.org/10.5281/zenodo.13363754).

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

## Acknowledgements

We thank Heather Wiedenhoft for assistance with zebrafish husbandry and the lateral line assays. We thank Dr. Hao Fan (Bioinformatics Institute, A*STAR Singapore) for the support. The authors would like to acknowledge the assistance given by the Research IT (Computational Shared Facility) of The University of Manchester, the National Supercomputing Center (NSCC) of Singapore and of the Bioinformatics Institute of A*STAR Singapore. A.B.'s Ph.D. fellowship was jointly funded by the University of Manchester and the Singapore ASTAR institute. This work was undertaken with the assistance of resources and services from the National Computational Infrastructure (NCI) and Pawsey Supercomputing Research Centre, which are supported by the Australian Government and the Government of Western Australia. M.L. O'Mara gratefully acknowledges funding from the Australian Research Council (ARC) (DP200100535). This work was supported by National Institute on Deafness and Other Communication Disorders (NIDCD) grants R15DC013900-01 (A.C.) and R01DC020701-01A1 (A.C.), National Institutes of Health (NIH) grants R01NS116043 (M.R.T.), and R01CA194057 (J.D.S.), R01CA194206(J.D.S.), P30 CA21745(St. Jude), NCI P30 CA021765 (St. Jude), and by the American Lebanese Syrian Associated Charities (ALSAC). R.C.K., M.M.S., and M.L. Oldham are part of Protein Technologies Center at St. Jude and supported by ALSAC. Md. Munan Shaik's current affiliation is the department of Structural Biology, Nimbus Therapeutics, Boston, MA, USA. Alexander Myasnikov's current affiliation is Dubochet Center for Imaging Lausanne (DCI-Lausanne), DCI CryoTEM Génopode EPFL, CH-1015 Lausanne, Switzerland. Jian Zuo's current affiliation is Ting Therapeutics, La Jolla, CA, USA.

## Author contributions

S.A.B., Y.F., and J.D.S. devised biochemical experiments and S.A.B., Y.F., J.L., and T.G. performed biochemical assays. K.A.W. and M.L. O'Mara devised the molecular dynamics simulations, K.A.W. performed the simulations and analyzed the data. M.M.S. and R.C.K. devised and troubleshooted purification protocol and structural determination experiments. M.M.S. developed large-scale protein purification strategy, expressed and purified protein, and optimized CryoEM conditions. R.C.K. assisted with protein expression and S.A.B assisted with protein purification. M.M.S. and A.M. collected CryoEM data, processed and refined the data to build the structural model and analyzed the structure. M.L. Oldham reprocessed the data and re-refined the model. A.B. and R.C.F assisted with structural analysis and developed the homology models of zebrafish Abcb6. R.A.U. and M.R.T. devised and performed WISH and MO injection experiments, and analyzed zebrafish morphants for the otolith phenotype. A.C. performed hair cell analyses. J.F. performed ABR testing and J.F, Y.W., and J. Zuo analyzed the data. Y.W. and J.F. isolated mouse cochlea for RNA-Seq. J. Zhu performed immunofluorescence microscopy experiments. S.A.B., K.A.W, A.C, M.L. O'Mara, and J.D.S compiled the manuscript with input from all authors.

## Competing interests

The authors declare no competing interests.
