## [Peer Review File · Nature Communications]

Reviewers' comments:

Reviewer #1 (Remarks to the Author):

In this article, the authors explored the potential relationship between ABCB6 and hearing. They reveal the unknown role of ABCB6 in inner ear development and function through cryoelectron microscopy studies, MD simulations, ABCB6 KO mice and auditory and vestibular studies in zebrafish. Overall, these substantial studies firmly establish the role of ABCB6 in the hearing system. Before the manuscript is considered further, several questions should be addressed.

(1): The L356P mutant was reported to abolish ABCB6 function by MD simulations based on WT cryoelectron microscopy structures. Therefore, validation of the current WT structure is important to support this conclusion. Please provide a flow chart of the cryo-EM data processing of WT ABCB6, and the local density of representative regions, around L356 residues.

(2): Lines 182-187, the authors simply superimpose the existing structure of ABCB6 with ATP based on the binding position of Sur1 and conclude that the binding mode of ATP and ABCB6 is in direct conflict. Is it possible to set up an MD simulation experiment to further explore the binding mode of the L356P mutant with ATP?

(3): Lines 222-224, can the ABCB6 structure of zebrafish be obtained by alpha fold server? If so, how does the predicted alpha fold structure differ from the current homologous modeled zebrafish ABCB6 structure?

(4): Line 248, data not shown; please show data.

(5): Please replace "uM" and "uL" with " μ M" and " μ L".

(6): In the supplemental table, the WT ABCB6 structure has a higher Clash Score and the Ramachandran plot has a lower preference value; please try to modify these cryoelectron microscopy parameters if possible.

(7): In Figure 5 and Supplementary Table 3, what is the phenotype of injected human L356P mRNA compared to wt human ABCB6 mRNA?

Reviewer #2 (Remarks to the Author):

The manuscript addresses the role of ABCB6 in hearing through the study of the impact of dyschromotosis universalis hereditaria (DUH)-associated mutations in the ABCB6 transporter using HEK293 cells. The DUH3 disease condition is linked to hearing loss. Analysis of seven ABCB6 mutants showed that only one of them, L356P, had an impact on ABCB6 expression in HEK293 and ATP hydrolysis. The authors obtained a high-resolution structure of ABCB6 using cryo-electron microscopy (EM) and used this structure to carry out molecular dynamics (MD) simulations on wild-type and the L356P mutant to understand the alteration caused by the mutation in the structure of ABCB6. In the second part of the study, the authors investigated the effect of knock down of *Abcb6* on zebra fish and in a mouse model. The zebrafish *Abcb6* morphants and knock-out mice experiments provide circumstantial evidence for *Abcb6*'s role in the development of the inner ear and hearing. One major concern about this study is that the cryoEM structure of ABCB6 and the characterization of seven mutations linked to DUH3 is an independent story that is not required for the most interesting and novel in vivo zebrafish morphants and knock-out mouse studies showing the role of ABCB6 in the development of inner ear and hearing function. However, the current in vivo findings provide only circumstantial evidence so additional experimental data is required on the localization of ABCB6 in zebra fish and mouse tissues including inner ear and some evidence is needed to determine whether the transport function of ABCB6 is required or it functions as part of a regulatory process along with other ABC or SLC transporter or cytoplasmic protein(s). We suggest the authors should focus in this manuscript only on the in vivo zebrafish morphants and mouse *Abcb6* knockout by providing additional data on the localization of *Abcb6* in the inner ear cells and whether the transport function is required by expressing non-functional (the Walker A, K to M or the Walker B, E to Q mutants) *Abcb6* in zebrafish or in a mouse model.

Specific comments:

1. The characterization of seven mutants was performed by assessing the binding to ATP agarose and hemin agarose using the cell lysates of HEK293 cells. There is no data provided on whether the ABCB6 protein in cell lysates was solubilized with a detergent. The level of expression of ABCB6 wild type and mutants in HEK293 might be in the range of 0.01 to 0.05% of total protein in cell lysates. It is not clear how the binding of other ATP-binding proteins (ATPases) to ATP-agarose was separated from that of ABCB6.

2. Other groups have already published the cryoEM structure of ABCB6. The authors have not provided any data or justification about how their structure adds any new information. They could have used published structures for the MD simulations studies.

3. For the MD simulations (Figure 2 D-N), did the authors use a homodimer of ABCB6? In that case, did they examine the changes caused by the L356P mutation in both monomers? For most homodimers of ABC transporters, the interaction of residues in both monomers is not identical. It would be helpful to look at the changes caused by the mutation in both monomers.

4. Zebrafish *Abcb6* morphants exhibited altered swimming behaviour, lateral line development, and utricular otoliths. Similar results were obtained using p53-null background zebrafish, a proper control to exclude morpholino p53-mediated off-target effects. Interestingly, the phenotype could only be rescued in part (53%) with human ABCB6 mRNA. This might suggest that the observed phenotype is not due to an *abcb6* homodimer, but by association with another membrane or cytoplasmic protein. This view is supported by the observation that individuals lacking the ABCB6 gene from the Lan blood group exhibit no obvious abnormalities, and heterozygous DUH individuals show a more pronounced phenotype than *abcb6*-null individuals. Is there any compensation for *Abcb6* knockout through the upregulation of another ABC or SLC transporter? The SHIELD database might provide some information.

5. Overall, the in vivo data suggest that *Abcb6* is required for normal inner ear development in zebrafish. But the main question of this study remains unanswered: what is the role of *Abcb6* in the inner ear development? The *Abcb6* knock-out mice exhibit reduced hearing sensitivity. The localization of *Abcb6* in inner ear cells and addressing whether the transport function of ABCB6 is required by expressing non-functional (the Walker A, K to M or the Walker B, E to Q mutants) *Abcb6* in zebrafish or in a mouse model.

6. Minor comments:

i. What is the role of the TMD0 domain in the function of ABCB6? Is the TMD0 domain involved in the stability and trafficking of ABCB6? The authors have not provided any information on the TMD0 domain of the wild type or the mutant ABCB6.

ii. What is the role of the N-linked glycosylation in human ABCB6? Is the localization of ABCB6 at the cell surface dependent on the N-linked glycosylation?

iii. Why was the ATP-binding position of Sur1 (ABCC8) (Fig. 2M) used? Could the authors use the same domain from another ABCB sub-family member (ABCB10)?

iv. In Figure 2E, the ATPase activity of the E-Q mutant of ABCB6 appears to be the same as the wild type at lower ATP concentrations. The lipid environment has a great influence on the ATPase activity of ABC transporters. Why were lipids not used along with the detergent to measure ATP hydrolysis?

Reviewer #3 (Remarks to the Author):

The manuscript of Baril et al., the result of a multinational collaboration, describes the characterisation of human ABCB6 mutants associated with dyschromatosis universalis hereditaria. Heterologous expression studies with seven likely causative mutants suggested that L356P, located in a coupling helix which links the transmembrane domain to the nucleotide binding domain, has the most debilitating effect on the protein. The structure of the wild-type transporter was solved by cryoEM (essentially confirming prior recent reports by others) and the L356P mutant was modelled into the coordinates. Molecular dynamic simulations suggested that the mutant protein was less flexible than the wild-type and that ATP binding would be sterically hindered, corroborating their biochemical data.

The major advance in the manuscript comes from the characterisation of Abcb6 in animal models; zebrafish Abcb6 was shown to be expressed in the otic vesicle consistent with a functional role in inner ear development. Interference of Abcb6 translation showed a reduction in utricular hair cells in the developing fish (knockdown of Abcb6 at the protein level was not verified). Importantly, rescue of the knockdown was achieved by co-injection of human ABCB6 mRNA. Behavioural changes in the Abcb6-knockdown animals were also observed and related to impaired inner ear function (the mRNA-injected rescued animals were not tested for concomitant rescue of this behaviour).

Hearing loss was then further confirmed in Abcb6-ko mice. Curiously, it is noted that ABCB6-null humans have no apparent phenotype. Possible explanations for this are explored.

Together, this is an interesting manuscript describing a new and important role for ABCB6 in the inner ear that will have implications for the diagnosis of patients and perhaps open the door to future treatments.

The manuscript however is not without its limitations and some data sets need tightening. I would also recommend that the scientific language is improved.

Comments for the authors to consider:

Line 26 onwards: the use of possessive apostrophes for gene/protein names and diseases is not good scientific language and, for me, grates throughout the manuscript. I recommend that all instances are rephrased (unless, of course, the journal finds this acceptable).

Line 40: please rephrase as not all in the ABC family function as ATP-dependent transporters

Line 42: yet ABCB6 has four TMDs – please clarify

Line 45: please rephrase as it could be interpreted that ABCB6 is the only ABC transporter implicated in multiple diseases which it is not

Line 61: please rephrase, macules do not have learning difficulties!

Line 93 to 95: If the mutations have been 'chosen' for the study then there is no justification for mentioning 'clustering'. Are these all the DUH mutants described? Please be clear.

Line 97: consider reporting the percentage identities of the homologs presented so it is easier to judge whether the regions in question are particularly conserved.

Line 104: are the western data in figure 1 just n=1? Is this reproducible?

Line 121: as above. There are no biological repeats, no statistical analyses of the data in figure 2a-d

Line 156: A comment on the absence of TMD0 from the cryoEM structure would be appropriate, especially as it was solved for pdb: 7D7N

Line 217: it seems a big assumption to suggest that lack of glycosylation of zebrafish Abcb6 will prevent expression at the plasmamembrane. You would need to prove this.

Line 260 and 343: please remove 'strongly' as rescue is somewhat variable

Line 450: were the hemin agarose beads washed prior to elution?

Line 686: please provide the number of biological repeats for figure 2e

Line 688: are these biological or technical repeats and the bracket is not closed

Line 719: is this number of replicates over different biological knockdown experiments. Please define

Figure 4: please consider including an image of the raw data for the hair cell count in treated v untreated zebrafish.

Figure 4: Would it be possible to include confirmation that Abcb6 protein expression has been reduced?

Figure 5: I'm a little confused by this experiment. I consider a rescue experiment critical and I am impressed that this has been tested because RNA interference is prone to off-target effects. Here, injection of human ABCB6 mRNA has been used to nullify the morpholino RNA. Why was zebrafish mRNA not used. Is there any possibility that the mRNA simply soaks up the morpholino and therefore it cannot act – (with siRNA experiments targeting message translation the best control would be to use a message in which the binding site for the siRNA has been modified silently; is the difficulty here simply because the morpholino targets a splice donor junction in the immature RNA. A brief description of this technique might suffice.

Reviewers' comments:

Reviewer #1:

(1): The L356P mutant was reported to abolish ABCB6 function by MD simulations based on WT cryoelectron microscopy structures. Therefore, validation of the current WT structure is important to support this conclusion. Please provide a flow chart of the cryo-EM data processing of WT ABCB6, and the local density of representative regions, around L356 residues.

Thank you for this suggestion. To support our MD simulation, we would like to point out that L356P, unlike WT ABCB6 is no longer robustly stabilized by nucleotide or exhibits a strong ATP dependent increase in catalytic activity. Thus, our MD simulations provide structural insights into why this might occur. The requested information has been included in the updated submission in supplementary figure 2 (flowchart) and supplementary figure 3 and 4 (local densities).

(2): Lines 182-187, the authors simply superimpose the existing structure of ABCB6 with ATP based on the binding position of Sur1 and conclude that the binding mode of ATP and ABCB6 is in direct conflict. Is it possible to set up an MD simulation experiment to further explore the binding mode of the L356P mutant with ATP?

As a point of clarification, we now use ABCB10 instead of ABCC8. Nonetheless this is a great question! To set up an MD simulation in which ATP is bound to the nucleotide binding site of ABCB6 would require ATP to first be docked to the site. As ABCB6 binds ATP with millimolar affinity (i.e., relatively low), this binding mode is highly conserved across all ABC transporters, and the conformation cannot be docked into the binding site due to the L356P mutation, it is not feasible to examine the interaction of ATP with a binding site that our empirical data shows nucleotide cannot physically bind to.

(3): Lines 222-224, can the ABCB6 structure of zebrafish be obtained by alpha fold server? If so, how does the predicted alpha fold structure differ from the current homologous modeled zebrafish ABCB6 structure?

Thank you for this helpful suggestion. We have conducted a comparison to the AlphaFold ABCB6 structure and show it is highly similar to our zebrafish homology model. This is now included in the manuscript in Supplementary Fig. 8.

(4): Line 248, data not shown; please show data.

This data has been added and can be found in Supplemental Figure 9.

(5): Please replace "uM" and "uL" with "µM" and "µL".

These changes have been included throughout the newest submission.

(6): In the supplemental table, the WT ABCB6 structure has a higher Clash Score and the Ramachandran plot has a lower preference value; please try to modify these cryoelectron microscopy parameters if possible.

*To address this our cryo-EM dataset has been re-processed with cryoSPARC, resulting in a higher resolution density map than our original result (now 2.93 Å vs. our previous 3.5 Å). The atomic model has been revised and all the validation statistics reflect these improvements (**Supplementary Table 2**). The new atomic model is in agreement with our original model (heavy atom RMSD = 1.56 Å) and the L356P sidechain positioning is not significantly altered, despite the revision. More details of the models along with a comparison with the original structure after equilibration MD simulations can be found below. The final model parameters can be found in the new updated supplementary figure 2-4 and supplementary table 2.*

(7): In Figure 5 and Supplementary Table 3, what is the phenotype of injected human L356P mRNA compared to wt human ABCB6 mRNA?

We cannot comment on the phenotype of a morpholino + L356P mRNA experiment. The chromatogram of L356P expressed in the absence of the stabilizing small molecule chaperone, 4-PBA is largely aggregated and seems misfolded L356P. This knowledge led to transfection of cells with an L356P- ABCB6. We showed that expression of L356P increased phosphorylation of eIF2 α which supports the proposition that a misfolded ABCB6 is activating the misfolded protein stress response (Supplementary Fig. 1d). In addition, attempts to produce stable L3356P-expressing cell lines failed because the cells did not tolerate L356P expression well and grew very slowly. It was for this reason that we did not attempt to inject L356P mRNA into zebrafish.

Reviewer #2:

1. The characterization of seven mutants was performed by assessing the binding to ATP agarose and hemin agarose using the cell lysates of HEK293 cells. There is no data provided on whether the ABCB6 protein in cell lysates was solubilized with a detergent. The level of expression of ABCB6 wild type and mutants in HEK293 might be in the range of 0.01 to 0.05% of total protein in cell lysates. It is not clear how the binding of other ATP-binding proteins (ATPases) to ATP-agarose was separated from that of ABCB6.

Other ATP-binding proteins are presumably bound to the ATP-agarose beads. However, the figure shows a Western Blot using an Anti-ABCB6 antibody, so only ABCB6 is visualized. The captions of this figure have been elaborated to prevent this misunderstanding in the future. The methods section reflects the lysis of ABCB6 in M-PER™ Mammalian Protein Extraction Reagent, a proprietary membrane protein stabilizing reagent available from ThermoFisher.

2. Other groups have already published the cryoEM structure of ABCB6. The authors have not provided any data or justification about how their structure adds any new information. They could have used published structures for the MD simulations studies.

When we started this project, there was no structure of ABCB6. During our investigation, other structures were published, some of which only purified ABCB6 lacking TMD0. However, our original structure obtained from purifying full length ABCB6 was the highest resolution ABCB6 structure at 3.5 Å. Therefore, we decided to use our structure to for MD simulations and determine if it could provide further insights into mutant ABCB6 non-functionality. Based on reviewer comments, we enlisted a new collaborator, Michael Oldham, to extensively reprocess the dataset with CryoSPARC, resulting in a 2.93 Å structure, a final structure that appears to be one of the best Cryo-EM of an inward-facing ABC transporter in the Protein Data Bank, especially considering the local resolution achieved in the transmembrane region.

3. For the MD simulations (Figure 2 D-N), did the authors use a homodimer of ABCB6? In that case, did they examine the changes caused by the L356P mutation in both monomers? For most homodimers of ABC transporters, the interaction of residues in both monomers is not identical. It would be helpful to look at the changes caused by the mutation in both monomers.

An ABCB6 homodimer was used in all MD simulations and the caption for Figure 2 has been updated to address this comment. We have updated our calculations in Supplementary Fig. 5 and 7 to include the single monomer analysis as requested

4. Zebrafish *Abcb6* morphants exhibited altered swimming behaviour, lateral line development, and utricular otoliths. Similar results were obtained using p53-null background zebrafish, a proper control to exclude morpholino p53-mediated off-target effects. Interestingly, the phenotype could only be rescued in part (53%) with human ABCB6 mRNA. This might suggest that the observed phenotype is not due to an *abcb6* homodimer, but by association with another membrane or cytoplasmic protein. This view is supported by the observation that individuals lacking the ABCB6 gene from the Lan blood group exhibit no obvious abnormalities, and heterozygous DUH individuals show a more pronounced phenotype than *abcb6*-null individuals. Is there any compensation for *Abcb6* knockout through the upregulation of another ABC or SLC transporter? The SHIELD database might provide some information.

*Although there is not 100% rescue, this is a reasonable value for a morpholino rescue experiment, especially when the morpholinos are directed at a splice junction and not interrupting translation. There are further explanations for the less than anticipated rescue: RNA injected at the one-cell stage gets diluted with each cell division and is degraded over time thus reducing the RNA available for rescue. The dose of RNA and morpholino is important too as too much morpholino can produce off-target effects like ocular coloboma. We empirically determined an amount of morpholino to inject that would not produce this effect. Notably, a 2014 study using morpholinos to knock down *pak4*, a serine/threonine protein kinase, reported only partial rescue (Law, S. H. W. & Sargent, T. D. PLOS ONE 9, e100268, doi:10.1371/journal.pone.0100268 (2014)).*

*Morpholino rescue percentages aside, we do not believe there is a heterodimer formed with ABCB6 to compensate for ABCB6 knockout (There was a BioRxiv paper that alluded an ABCB6:ABCB5 heterodimer, however we could not detect ABCB5 in cochlea). Further, the SHIELD database shows that many ABC transporters are not abundant in the inner ear (Supplementary Figure 11) during development, which means there are fewer potential players for either compensation or physical interaction. This observation is not surprising, given that the inner ear is a tightly regulated space, purposefully cut off from the blood supply to prevent damaging compounds from accessing the sensitive structures required for hearing and balance. Of the ABCB subfamily, only the mitochondrial transporters *Abcb6*, *Abcb7* *Abcb8*, and *Abcb10* show detectable read counts compared to the other ABCB proteins. To gain insight into ABCB6 absence, we have included new RNA-seq analysis of the WT and *Abcb6**

knockout mouse cochlea. We did not observe any differences in expression of these ABC transporters compared to WT in the Abcb6 knockout cochlea. However, our mouse studies were performed with cochlea isolated from 2–3-month mice, when hearing is fully mature and before any age-related hearing loss.

5. Overall, the in vivo data suggest that Abcb6 is required for normal inner ear development in zebrafish. But the main question of this study remains unanswered: what is the role of Abcb6 in the inner ear development? The Abcb6 knock-out mice exhibit reduced hearing sensitivity. The localization of Abcb6 in inner ear cells and addressing whether the transport function of ABCB6 is required by expressing non-functional (the Walker A, K to M or the Walker B, E to Q mutants) Abcb6 in zebrafish or in a mouse model.

We agree that reviewer's question is important. Based on the SHIELD data, Abcb6 is expressed in the mouse cochlea (Fig. 7a) and because of this we performed new studies where we obtained data from an RNA seq analysis of murine cochlea from ABCB6 WT and KO mice. Our new supplemental data shows that the absence of ABCB6 while not affecting other ABCB family members does impact the expression of genes involved in hearing. Some of these findings align with the additional new data from zebrafish showing knockdown of ABCB6 reduces the number of hair cells in the macula. Future studies will require the development of ABCB6 mice with a conditional allele that will permit deletion of ABCB6 at specific developmental time points. Such a mouse takes time to develop and should ultimately permit experiments that reveal more clearly ABCB6 contribution to the complex process of inner ear development (however answering this question would be a publication in and of itself).

6. Minor comments:

i. What is the role of the TMD0 domain in the function of ABCB6? Is the TMD0 domain involved in the stability and trafficking of ABCB6? The authors have not provided any information on the TMD0 domain of the wild type or the mutant ABCB6.

The TMD₀ is a well-known disordered domain, has been elusive in all published structures, except one low resolution structure. In most of the published structures in detergent and nanodiscs, including our own, density for the TMD₀ is missing from the final model. In our preparation of ABCB6 in DDM/CHS, the TMD₀ is too flexible to emerge in the 2D class projections. This observation is consistent with a previous structural determination of ABCB6 bound to hemin and glutathione (PDB 7DNZ) in the presence of Cymal-6/CHS, another detergent system. A previous 5.20 Å reconstruction of ABCB6 in the presence of LMNG (PDB 7D7N), showed the TMD₀ in the 2D class projections with density in the final reconstruction, however the resulting structure is 5.2 Å. This low resolution is not sufficient to allow confident MD simulations of the full length ABCB6. Moreover, in later publications the authors showed that the internal cavity was populated by two internal loops, not addressed in 7D7N. We also observed the presence of the TMD₀ in 2D class projections from a similar low resolution reconstruction using LMNG in the absence of CHS (data not shown) but chose our higher-resolution

structure resulting from the DDM/CHS detergent system for MD. Therefore, we cannot comment on the role of TMD0 in either the WT or L356P mutant, except to say the latter poorly binds nucleotide which is consistent with its minimal ATPase activity.

ii. What is the role of the N-linked glycosylation in human ABCB6? Is the localization of ABCB6 at the cell surface dependent on the N-linked glycosylation?

*ABCB6 harbors a single-non-consensus glycosylation site (NXC) in TMD0 (Fukuda et al **JBC** 2011). N-linked glycosylation broadly allows for trafficking to the plasma membrane through secretory transport. The literature presents conflicting accounts of how glycosylation affects ABCB6, however. One group found that the non-glycosylated form of ABCB6 trafficked intracellularly while the plasma membrane was enriched for a glycosylated form of ABCB6 (Paterson et al, *Biochemistry* **2007**, 46, 9443-9452). A second group (lead by one of the authors from the *Biochemistry* 2007 paper, had opposing results: when the domain containing the N-linked glycosylation site was removed (TMD₀ domain), the rest of the protein trafficked to the plasma membrane, while the full-length protein resided intracellularly (Kiss et al, *PLoS One* **2012**; 7(5):e37378.), although there was no discussion of glycosylation of the protein. A recent study of a paralog of ABCB6 (HMT-1) shows that deletion of TMD0 (the authors refer to it as NTE, N-terminal Extension) resulted in mistargeting to the plasma membrane. On balance the TMD0 and N-linked glycosylation are probably key to trafficking.*

iii. Why was the ATP-binding position of Sur1 (ABCC8) (Fig. 2M) used? Could the authors use the same domain from another ABCB sub-family member (ABCB10)?

Great question. We used it because of the conservation of the nucleotide binding domain. However, in the updated submission we show the ATP position from ABCB10 (see Figure 2j-l for the updated figure). A comparison indicates that the results are very similar if we use ABCB10 instead of ABCC8.

iv. In Figure 2E, the ATPase activity of the E-Q mutant of ABCB6 appears to be the same as the wild type at lower ATP concentrations. The lipid environment has a great influence on the ATPase activity of ABC transporters. Why were lipids not used along with the detergent to measure ATP hydrolysis?

*All of the ABCB6 proteins used were purified in the presence of both n-dodecyl- β -maltoside (DDM) and cholesteryl hemisuccinate (CHS), a cholesterol ester or cholesterol lipid, as is common in other purification protocols. ATPase assays were performed according to a commonly accepted protocol developed by Chifflet et al (Chifflet, S et al. *Analytical Biochemistry* **168**, 1-4, (1988)) for measuring ATP-ase activity. This protocol has been referenced 424 times according to Scopus with recent citations in vhigh-impact journals using this technique with other membrane transporters, including ABCG2, ABCB1, and ABCB6. Because of the reviewer's concern, we purified new proteins based on re-examination of our graphs. We agree*

with the reviewer that the E752Q mutant did appear to have some activity at lower concentrations. Since another reviewer had raised an issue with the reproducibility of our ATPase data, we performed additional experiments that have been included in the updated version. In addition, we performed a series of E752Q assays. When looking at the data, we noticed a data point in the original data set that seemed very high for the E752Q dataset. The datapoint was calculated to be an outlier using Prism, and excluded from the new, complete dataset included in Figure 2C. We also have plotted these graph on a log2 scale so that all datapoints are visible. The breakdown of the E752Q ATPase data can be found below. The explanation for the previous findings appears to be because of the variation in ATPase activity at the low ATP concentration not something intrinsic.

Reviewer #3 (Remarks to the Author):

Comments for the authors to consider:

Line 26 onwards: the use of possessive apostrophes for gene/protein names and diseases is not good scientific language and, for me, grates throughout the manuscript. I recommend that all instances are rephrased (unless, of course, the journal finds this acceptable).

This comment has been addressed in the most current submission.

Line 40: please rephrase as not all in the ABC family function as ATP-dependent transporters

This comment has been addressed in the most vcurrent submission.

Line 42: yet ABCB6 has four TMDs – please clarify

This comment has been addressed in the most current submission.

Line 45: please rephrase as it could be interpreted that ABCB6 is the only ABC transporter implicated in multiple diseases which it is not

This comment has been addressed in the current submission.

Line 61: please rephrase, macules do not have learning difficulties!

This comment has been addressed in the most recent submission.

Line 93 to 95: If the mutations have been ‘chosen’ for the study then there is no justification for mentioning ‘clustering’. Are these all the DUH mutants described? Please be clear.

All the reported DUH missense mutations (as of the start of the study) were selected for investigation, except for the incorrectly identified S322K, which was correctly identified as S322R. The text has been updated to reflect this information in the ‘Expression of DUH Mutants’ section within the Results.

Line 97: consider reporting the percentage identities of the homologs presented so it is easier to judge whether the regions in question are particularly conserved.

Please see the caption of Figure 1 as well as the ‘Sequencing Alignments’ section within Materials and Methods.

Line 104: are the western data in figure 1 just n=1? Is this reproducible?

Representative western blots were included in Figure 1; however we have updated the quantitative data in Figure 1 c (N=5) and f (N=3 for hemin agarose and N=4 for ATP agarose) as well as the figure captions to make this clearer.

Line 121: as above. There are no biological repeats, no statistical analyses of the data in figure 2a-d

Thank you for the comment. This was a mistake. In Figure 2 a-d and the corresponding caption we updated the datasets. Representative data is shown for the Western blots in a and d, however statistical information is now included for b, c, and e.

Line 156: A comment on the absence of TMD0 from the cryoEM structure would be appropriate, especially as it was solved for pdb: 7D7N

Please see 'Structural Insights into L356P from Cryo-EM and MD Simulations within the Results section. We have commented on the TMD0 as requested by the reviewer; however, vabf vs bwe do remind the reviewer that other publications have either removed the TMD0 entirely from their expression construct or have not observed density from this domain in their resulting structure.

Line 217: it seems a big assumption to suggest that lack of glycosylation of zebrafish Abcb6 will prevent expression at the plasma membrane. You would need to prove this.

Please see supplementary figure 6 e-g for microscopy images showing that zebrafish Abcb6 is located intracellularly.

Line 260 and 343: please remove 'strongly' as rescue is somewhat variable

This comment has been addressed in the most recent submission.

Line 450: were the hemin agarose beads washed prior to elution?

Yes, all the pulldowns featured 3 wash steps prior to elution. We have updated the methods to reflect this comment.

Line 686: please provide the number of biological repeats for figure 2e

Thank you. This comment has been addressed in the most recent submission in the caption for figure 2e (now figure 2 c)

Line 688: are these biological or technical repeats and the bracket is not closed

This comment has been addressed in the most recent submission to reflect that they are biological replicates.

Line 719: is this number of replicates over different biological knockdown experiments. Please define. vvv

The sample size indicates the number of fish used for the experiment shown here. This experiment was repeated, and we obtained similar results.

Figure 4: please consider including an image of the raw data for the hair cell count in treated v untreated zebrafish.

We unfortunately do not have the raw images. Hair cells are counted while looking through a compound microscope, not from images, and we no longer have the fish since it was years ago.

Figure 4: Would it be possible to include confirmation that Abcb6 protein expression has been reduced?

Unfortunately, we have not found an antibody that would react in zebrafish tissues to demonstrate that Abcb6 expression has been reduced.
ccurrentv

Figure 5: I'm a little confused by this experiment. I consider a rescue experiment critical and I am impressed that this has been tested because RNA interference is prone to off-target effects. Here, injection of human ABCB6 mRNA has been used to nullify the morpholino RNA. Why was zebrafish mRNA not used? Is there any possibility that the mRNA simply soaks up the morpholino and therefore it cannot act – (with siRNA experiments targeting message translation the best control would be to use a message in which the binding site for the siRNA has been modified silently; is the difficulty here simply because the morpholino targets a splice donor junction in the immature RNA. A brief description of this technique might suffice.

We agree that rescue experiments are important. After we discovered hearing deficits in the mouse and similar vestibular defects in zebrafish (and that zebrafish ABCB6 was structurally similar to mammalian ABCB6) we wanted to rescue the zebrafish phenotype with a mammalian ortholog. Our reasoning was as follows: Despite their homology, the zebrafish Abcb6 morpholino oligo seemed unlikely to bind the human mRNA, thus it was unlikely that the mRNA would “soak up” the morpholino so that the morpholino cannot properly bind to the splice junction. However, if the zebrafish Abcb6 mRNA was injected, the zebrafish mRNA contains some morpholino binding sites and so the morpholino oligo would bind both the endogenous and injected zebrafish Abcb6 mRNA (which is comparable to the “soaking up” scenario the reviewer proposed was happening with the human mRNA).

REVIEWER COMMENTS

Reviewer #1 (Remarks to the Author):

The authors have significantly improved the manuscript through revision and responded well to my comments. I have no further concerns.

Reviewer #3 (Remarks to the Author):

I am pleased to recommend publication of Baril et al., in Nat Comms. The manuscript describes an eclectic approach to the study of this transporter to derive important conclusions relating to the role of AbcB6 in the inner ear. My concerns relating to the original manuscript have been fully addressed.

Reviewer #4 (Remarks to the Author):

This is an important manuscript. I was not an original reviewer but was asked to look at v2. With a fresh pair of eyes it is clear that the revision of the manuscript deals with the concerns raised by reviewers during the initial submission of the manuscript. The inclusion of new RNAseq data, further cryo-EM processing and in vitro protein biochemistry strengthens the manuscript considerably. To undertake all this additional experimentation is commendable. It would be churlish to quibble too much over any of the data further, but I do have one question and one comment.

Question: The ATPase activity data was obviously a source of concern in the original manuscript. The new data on the Walker-B mutation (E752Q) shows clearly this is a null mutant (although actually the original data wouldn't have bothered me) and that L536P is also essentially unable to hydrolyse ATP. What is odd is the WT which doesn't show anything like a Michaelis-Menten plot for "basal" ATPase activity. Perhaps this is due to the detergent solubilisation state of the protein. I wonder if the authors could include a comment on this in the manuscript.

Comment: Figure 6f should have time on the y-axis

Reviewer #5 (Remarks to the Author):

Lines 238-271: Within the supplemental information, please include data confirming 1) *abcb6* knockdown following injection of the splice-blocking *Abcb6* morpholino and 2) the absence of *abcb6* knockdown following injection of water alone or the scrambled *Abcb6* morpholino. These data are critical for supporting all of the conclusions within this section, as the phenotypes observed may be an artifact of injection stress and/or off-target effects of the *Abcb6* morpholino. Once these data are included, please update this section to summarize your results based on confirmation of knockdown.

Line 636: Please indicate whether you used splice-blocking or translational-blocking morpholinos. This is important since the approach to confirming knockdown is different depending on the type of morpholino used. In addition, more details are needed about 1) the buffer and concentration of morpholino stocks; 2) the make/model of the microinjection and micromanipulator systems used; and 3) the injection needles, injection volume, and injection parameters used for microinjection. This information is critical to ensure that other labs can replicate your work.

Figures 4-6: Please include data for water-only injected embryos and embryos injected with scrambled *Abcb6* morpholino across all treatments. These negative controls are critical for confirming that phenotypes within morphants are not an artifact of injection stress and/or off-target effects of the morpholino.

Response to reviewers

You will see that, while Reviewers #1 and #3 sign off, and Reviewer #4 only has one comment, Reviewer #5 has noted several places where further data is required for the morpholino experiments.

With respect to reviewer #4

Reviewer #4 (Remarks to the Author):

This is an important manuscript. I was not an original reviewer but was asked to look at v2. With a fresh pair of eyes it is clear that the revision of the manuscript deals with the concerns raised by reviewers during the initial submission of the manuscript. The inclusion of new RNAseq data, further cryo-EM processing and in vitro protein biochemistry strengthens the manuscript considerably. To undertake all this additional experimentation is commendable. It would be churlish to quibble too much over any of the data further, but I do have one question and one comment.

Question: The ATPase activity data was obviously a source of concern in the original manuscript. The new data on the Walker-B mutation (E752Q) shows clearly this is a null mutant (although actually the original data wouldn't have bothered me) and that L536P is also essentially unable to hydrolyse ATP. What is odd is the WT which doesn't show anything like a Michaelis-Menten plot for "basal" ATPase activity. Perhaps this is due to the detergent solubilisation state of the protein. I wonder if the authors could include a comment on this in the manuscript.

*Response: Thank you for your positive overall assessment of our manuscript and the opportunity to address this issue. We previously had switched our graphs to plots with a log₂ x-axis in an attempt to show that the lower concentration points were visible. Likely, this way of displaying the data complicated the visual interpretation. However, when we switched the graph back to a linear x-axis, the shape of the curve still appeared to deviate from the normal Michaelis-Menton graph shape. We thought this might be due to elevated signal-to-noise ratio at low protein concentrations as well as batch-to-batch differences in the purified proteins. To investigate this possibility, we acquired more pure protein and then first assayed huABCB6 WT at both 5 µg and 25 µg of protein (See **a, below**). Although the v_{max} is similar between the two graphs, shape of the graph at low concentrations is slightly different, but the trend is similar at the two concentrations. Nonetheless, in all cases, the E752Q and L356P mutant curves remain very different from the ABCB6 WT curve, thus the the difference in the shape of the curve does not appear to affect our conclusions regarding the L356P mutant's decreased ability to bind ATP compared to ABCB6 WT. We have added the additional datasets for the 5 µg ABCB6 WT ATPase into our Figure 2c. The original graph is included below (b) as well as the updated graph (c). We believe that in the updated graph, the huABCB6 WT looks less "odd." Furthermore, many of our ATPase assays used protein that was used in our CryoEM experiments, so we are confident that the protein is properly solubilized in the micelle based on the observation of the micelle in the 2D classifications and ab initio refinements (see supplemental figure 2).*

Comment: Figure 6f should have time on the y-axis

Response: *This comment has been addressed in the updated manuscript.*

With respect to reviewer #5

Reviewer #5 (Remarks to the Author):

Lines 238-271: Within the supplemental information, please include data confirming 1) *abcb6* knockdown following injection of the splice-blocking *Abcb6* morpholino and 2) the absence of *abcb6* knockdown following injection of water alone or the scrambled *Abcb6* morpholino. These data are critical for supporting all of the conclusions within this section, as the phenotypes observed may be an artifact of injection stress and/or off-target effects of the *Abcb6* morpholino. Once these data are included, please update this section to summarize your results based on confirmation of knockdown.

Line 636: Please indicate whether you used splice-blocking or translational-blocking morpholinos. This is important since the approach to confirming knockdown is different depending on the type of morpholino used. In addition, more details are needed about 1) the buffer and concentration of morpholino stocks; 2) the make/model of the microinjection and micromanipulator systems used; and 3) the injection needles, injection volume, and injection parameters used for microinjection. This information is critical to ensure that other labs can replicate your work.

Figures 4-6: Please include data for water-only injected embryos and embryos injected with scrambled *Abcb6* morpholino across all treatments. These negative controls are critical for confirming that phenotypes within morphants are not an artifact of injection stress and/or off-target effects of the morpholino.

Additionally, please ensure that you follow the guidance included in Stainier et al. (2017) regarding morpholino experiments. In particular please ensure your experiments comply with the following points:

Response: As suggested we have used a second morpholino (MO7) and performed the corresponding RT-PCR to assess the efficiency of both MOs. Our original MO, which targeted the exon 15 splice junction, is now annotated in the text as MO15. In addition and as mentioned below, we used a commercially provided scrambled control morpholino.

- Rescue experiments should be attempted for the approaches listed above (e.g., by injection of mRNA or DNA lacking the MO-binding site in the case of MO studies), and if rescue is successful, control experiments should be conducted (e.g., using mutant RNA or DNA, i.e., RNA or DNA that does not encode a functional gene product).

*Response: We conducted an mRNA rescue experiment in embryos injected with MO15 and achieved a partial rescue, suggesting that our morphants have a specific knockdown of *abcb6* (fig. 5c, supplementary table 4). Although the rescue was not 100%, only 5.3% of fish co-injected with MO15 and *ABCB6* mRNA retained the phenotype observed in 95% of fish injected with MO15. 52.6% of fish co-injected with MO15 and *ABCB6* mRNA exhibited the WT phenotype with 2 saccules and 2 utricles and 42.1% of co-injected fish exhibited an intermediate phenotype with 2 saccules and 1 utricle. Please note that this partial rescue of MO knockdown is not unique to our experiments and has been commonly reported in other high quality peer-reviewed journals.*

- An injection control MO (standard negative control MO, 5-base mismatch MO, or a suitable alternative MO) should be used to account for developmental delay. Such MOs cannot serve as controls for the specificity of the experimental MO.

Response: A commercial scrambled control MO available from GeneTools LLC was used in this revision.

- The approach of validating ATG MOs by assessing their ability to suppress the expression of a co-injected target mRNA-GFP fusion is of little value as we now know that suppression of GFP expression is generally observed and it does not test the effect of the MO on the endogenous RNA. This control is no longer recommended.

Response: We agree and note this approach was not used in either the original manuscript or in this revision.

- Essential routine procedures include a dose-response curve (extra caution should be exercised when one has to inject more than 5 ng of a MO to cause a phenotype (Schulte-Merker et al., 2014), the examination of statistically meaningful numbers of control and experimental animals, extensive documentation of the penetrance and expressivity of all

phenotypes, and the use of blinding strategies whenever possible.

Response: In supplementary figure 12, we have included the results of our morpholino dose-response, both with images of the otolith and the corresponding RT-PCR quantitation of the single-otolith phenotype in the injected zebrafish (primary data in supplementary table 7). Lastly, we hasten mention that with exception of the defects we noted after Abcb6 knockdown, all images of the morphants indicate normal development.

References:

Stainier DYR, Raz E, Lawson ND, Ekker SC, Burdine RD, Eisen JS, et al. (2017) Guidelines for morpholino use in zebrafish. PLoS Genet 13(10): e1007000. <https://doi.org/10.1371/journal.pgen.1007000>

Schulte-Merker S, Stainier DY. Out with the old, in with the new: reassessing morpholino knockdowns in light of genome editing technology. Development. 2014;141:3103–4. pmid:25100652.

REVIEWERS' COMMENTS

Reviewer #4 (Remarks to the Author):

I had only 1 or 2 minor comments on the previous version of this manuscript and somewhat surprised to be asked to see it again. No further comments from me. The manuscript should be published.

Reviewer #5 (Remarks to the Author):

The authors have adequately addressed my prior comments. Recommend accepting for publication.